# The CONSTANS flowering complex controls the protective response of photosynthesis in the green alga *Chlamydomonas*

Ryutaro Tokutsu [1,2], Konomi Fujimura-Kamada[1], Takuya Matsuo[3], Tomohito Yamasaki [4] & Jun Minagawa [1,2]

Light is essential for photosynthesis, but the amounts of light that exceed an organism's assimilation capacity can result in oxidative stress and even cell death. Plants and microalgae have developed a photoprotective response mechanism, qE, that dissipates excess light energy as thermal energy. In the green alga *Chlamydomonas reinhardtii*, qE is regulated by light-inducible photoprotective proteins, but the pathway from light perception to qE is not fully understood. Here, we show that the transcription factors CONSTANS and Nuclear transcription Factor Ys (NF-Ys) form a complex that governs light-dependent photoprotective responses in *C. reinhardtii*. The qE responses do not occur in *CONSTANS* or *NF-Y* mutants. The signal from light perception to the CONSTANS/NF-Ys complex is directly inhibited by the SPA1/COP1-dependent E3 ubiquitin ligase. This negative regulation mediated by the E3 ubiquitin ligase and the CONSTANS/NF-Ys complex is common to photoprotective response in algal photosynthesis and flowering in plants.

[1] Division of Environmental Photobiology, National Institute for Basic Biology, Nishigo-naka 38, Myodaiji, Okazaki 444-8585, Japan. [2] Department of Basic Biology, School of Life Science, Graduate University for Advanced Studies, Okazaki 444-8585, Japan. [3] Center for Gene Research, Nagoya University, Nagoya 464-8602, Japan. [4] Science and Technology Department, Natural Science Cluster, Kochi University, 2-5-1 Akebono-cho, Kochi 780-8520, Japan. Correspondence and requests for materials should be addressed to R.T. (email: tokutsu@nibb.ac.jp) or to J.M. (email: minagawa@nibb.ac.jp)

Photosynthetic organisms are often exposed to excess light. This can result in severe oxidative stress and even cell death. Plants and algae have developed a photoprotective response mechanism, qE, which is driven by low pH in the thylakoid lumen generated by the photosynthetic electron flow[1]. This lumenal low pH modifies the light-harvesting complex II pigment composition via the xanthophyll cycle and activates specific photoprotective proteins, such as PSBS and LIGHT-HARVESTING COMPLEX STRESS-RELATED PROTEINS (LHCSRs)[2]. Mutants deficient in these effector proteins are highly stressed under high light (HL) conditions[3,4]. In a green alga Chlamydomonas reinhardtii, qE-type photoprotection is light-inducible via the expression of photoprotective proteins (LHCSRs and PSBS)[5]. Two photoreceptors—PHOTOTROPIN (PHOT)[6] and ultra-violet (UV) RESISTANCE LOCUS 8 (UVR8)[7]—were identified as initiation factors for signal transduction of the photoprotective genes and protein expression in the green alga. Subsequently, two *phot* suppressor loci were identified to be involved in the induction of *LHCSRs*: DE-ETIOLATED 1 (*det1*) and DAMAGED DNA-BINDING 1 (*ddb1*) and an E3 ubiquitin ligase complex, CUL4–DDB1$^{DET1}$, was proposed to mediate the PHOT signal to *LHCSRs* gene expression[8]. Despite these findings, we still do not know most of other important players in the signal transduction pathways, especially transcription factors that directly activate light-dependent gene expression. During a recent genetic screening using a bioluminescence reporter assay, however, we obtained several Deficient in LHCSR expression (*DSR*) mutants that showed reduced expression of *LHCSR* genes[9]. Two of the mutants, *DSR10* and *DSR15*, had a mutation in *CONSTANS* and two other mutants, *DSR28* and *CC4286*, had a mutation in *NF-YB*.

CONSTANS (CO) is a circadian clock-regulated gene encoding a transcription factor required for flowering. Since Coupland et al. first reported that CO mediates photoperiodic flowering by directly regulating the transcription of *FLOWERING LOCUS T* (*FT*)[10], extensive studies have revealed that CO and its ubiquitination under the control of photoreceptors are the central to regulation of photoperiodic flowering[11]. NUCLEAR FACTOR Y (NF-Y) proteins (NF-YA, NF-YB, and NF-YC) bind to the promoter regions of their target genes[12] and are widely conserved in eukaryotic organisms, including yeast, mammals, and plants. NF-Ys are the most thoroughly studied proteins that interact with CO[13]. The biological functions and underlying molecular mechanisms of NF-Y proteins have been intensively investigated in both animals and plants[14]. For example, Wenkel et al. reported that in flowering plants, HEME ACTIVATOR PROTEIN, which corresponds to NF-YA, strongly interacts with CO, forming a transcriptional complex that regulates the expression of its downstream gene, *FT*[15]. The interaction of NF-YB2/3 and NF-YC3/4/9 with CO has physiological relevance in plants, as the absence of any of these proteins results in delayed flowering[13]. A comparable sophisticated mechanism has not been identified in microalgae, as their population growth relies on cell division, not on flowering as in land plants.

Even though C. reinhardtii does not produce flowers, a copy of each of these transcriptional factors has been identified on its genome, including *CONSTANS* (*CrCO*), *NF-YB*, and *NF-YC*[16,17]. Previously, heterologous expression of *CrCO* was shown to complement the function of CO in the flowering plant *Arabidopsis thaliana*[16]. Phylogenetic analysis of *CO* and *CO-like* (*COL*) genes from green algae and land plants indicated that *CrCO* is a precursor of *CO* in green photosynthetic eukaryotes[18]. The C. reinhardtii homologs of NF-YB and NF-YC were clustered with NF-YB1/8/10 and with NF-YC1/2/3/4/9, respectively, in phylogenetic analyses of the NF-Y protein family (Supplementary Fig. 1)[17]. Although few physiological and biochemical analyses of

algal NF-Ys have been reported to date[17], these findings suggest that the CO/NF-YB/NF-YC transcriptional complex might have arisen before the divergence of land plants. In the current study, using the *DSR* mutants defective in *CrCO* or *NF-YB* and other newly constructed mutants in C. reinhardtii, we show that the transcription factors CrCO and NF-Ys form a complex that governs light-dependent photoprotective responses in C. reinhardtii. Further, we show that the CrCO/NF-Ys complex was directly inhibited by the SUPPRESSOR OF PHYA-105 1 (SPA1) and CONSTITUTIVE PHOTOMORPHOGENIC 1 (COP1) -dependent E3 ubiquitin ligase. We propose that CrCO and its ubiquitination is the central to regulation of photoprotection in C. reinhardtii, much as like flowering in plants.

## Results

**CONSTANS and NF-YB are crucial for photoprotection.** We identified mutations in *CrCO* in two *DSR* mutants, *DSR10* (*crco-1*) and *DSR15* (*crco-2*) (Supplementary Fig. 2a Cre06.g278159). A combination of linkage mapping and whole-genome sequencing revealed that two other mutants, *DSR28* (*nfyb-1*) and *CC4286* (*nfyb-2*), had mutations in *NF-YB* (Supplementary Fig. 2a, Cre02.g079200). While *DSR10* (*crco-1*) and *DSR15* (*crco-2*) had reduced *CrCO* yet normal *NF-YB* transcript levels, *DSR28* (*nfyb-1*) and *CC4286* (*nfyb-2*) had normal *CrCO* yet reduced *NF-YB* transcript levels (Supplementary Fig. 2b). Similarly, RT-PCR analysis revealed that the transcripts of representative genes involved in photoprotection in C. reinhardtii (LHCSRs and PSBSs) were strongly reduced or almost undetectable in all of these mutants[9] (Supplementary Fig. 3). Under 30 µmol photons m$^{-2}$ s$^{-1}$ (low light [LL]) and 100 µmol photons m$^{-2}$ s$^{-1}$ (medium light), these mutants had comparable growth rates and similar cell shapes to the wild-type strains (Supplementary Fig. 4). We cannot exclude the possibility that the photoperiodic responses of these mutants were altered as described in the previous report[16]. However, the mutation in *CrCO* or *NF-YB* had little effect on the cell division process under low to medium light conditions. These findings suggest that either CrCO or NF-YB, or both, are crucial regulators of photoprotective responses in C. reinhardtii, and are therefore necessary for survival under HL conditions. This is an intriguing hypothesis, given that their homologs in land plants function together as a flowering module[11,13].

To test this hypothesis, we first measured photoprotective thermal dissipation of excess absorbed energy in the mutants. This process occurs via the quenching mechanism qE, which is the fastest reversible photoprotective response. It suppresses photo-oxidative damage and contributes to optimizing photosynthetic efficiency under variable light conditions in nature[19]. In C. reinhardtii, qE-type photoprotection is a light-inducible process that involves the expression of photoprotective proteins[6,7]. When *DSR15* (*crco-2*) and *DSR28* (*nfyb-1*) strains were exposed to excess light (including weak UV), they exhibited pigment bleaching (Fig. 1a, b). This is an indicator of severe photoinhibition and is accompanied by a decrease in the maximum quantum efficiency of photosystem II (Fv/Fm) (Fig. 1c). These mutations were successfully rescued via expression of either CrCO (*crco-2/CrCO*) or NF-YB (*nfyb-1/NFYB*) fused with a YFP variant, Venus–3xFLAG (Fig. 1a–d). The photoinhibitory phenotype observed in *crco-2* and *nfyb-1* may have been due to insufficient expression of the key photoprotective proteins, including LHCSR1, LHCSR3, and PSBS, because the mutants' qE activities were almost lost (Fig. 1d, e). These results indicate that both CrCO and NF-YB are essential for the functional activation of qE-dependent photoprotection in C. reinhardtii.

The kinetics of the photoprotective response of the rescued strains were then examined. The *crco-2/CrCO* strain showed a delayed response in both qE and photoprotective protein

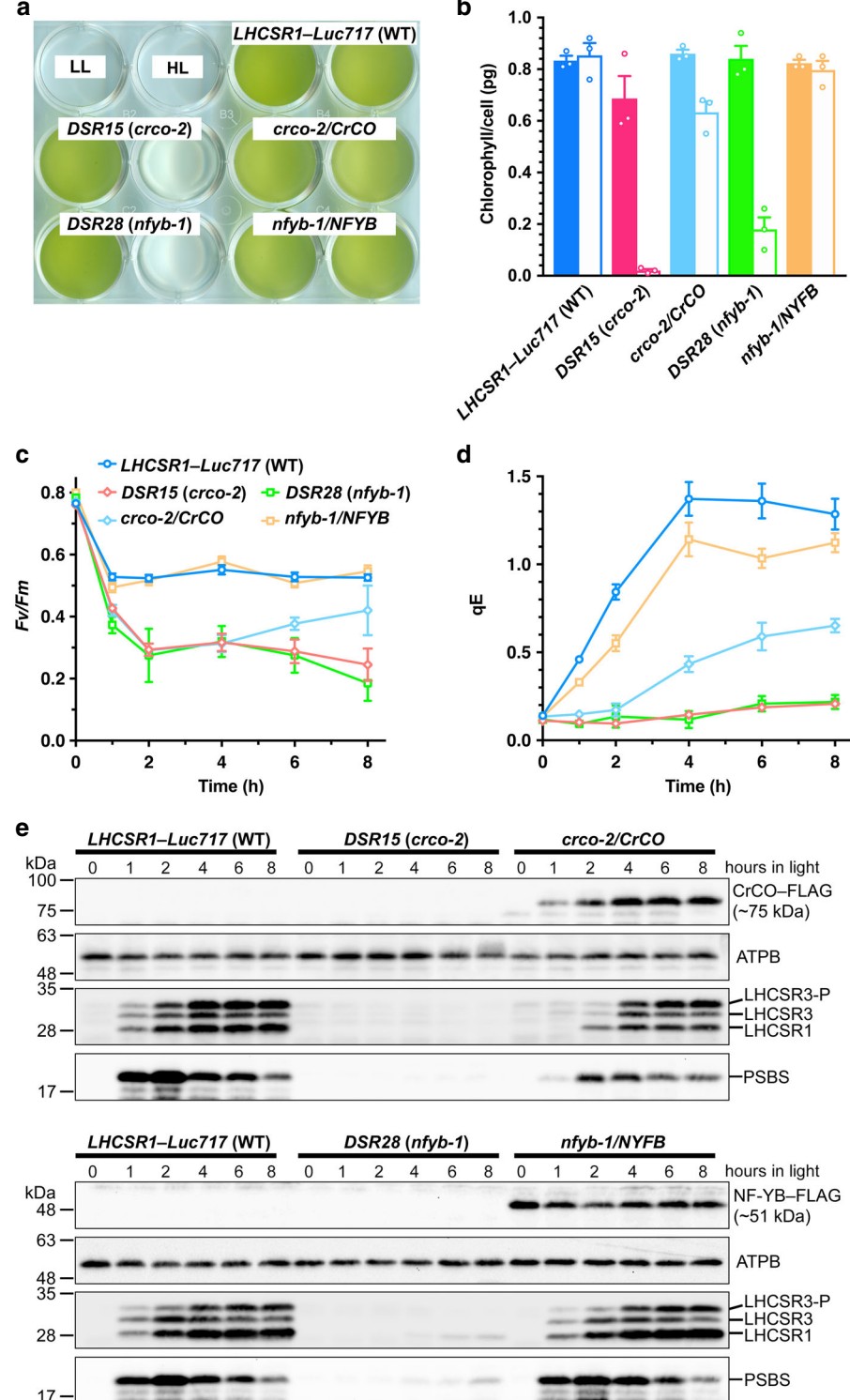

**Fig. 1** CrCO and NF-YB are crucial for photoprotection. **a** The bleaching phenotypes of the reference strain (*LHCSR1–Luc717*) and the *DSR* mutants visualized in multiwell plates. Representative cell cultures treated with low light (LL; left wells) or high light (HL; right wells). Concentrations of the cultures were adjusted to 1.0 × 10$^7$ cells/mL. **b** Chlorophyll content per cell after LL (closed bar) or HL (open bar) treatment of the cells shown in **a**. **c** Maximum quantum yield of photosystem II (Fv/Fm) during HL treatment. **d** qE quenching capability during HL treatment. **e** Immunoblot analysis of 3xFLAG-fused proteins (CrCO–FLAG and NF-YB–FLAG in *crco-2/CrCO* and *nfyb-1/NFYB*, respectively), LHCSR1, LHCSR3, and PSBS during HL treatment. ATPB protein levels are shown as the loading control. The experiments were performed three times with different biological samples (*n* = 3 biological replicates; mean ± S.E.M); a representative experiment is shown in **e**

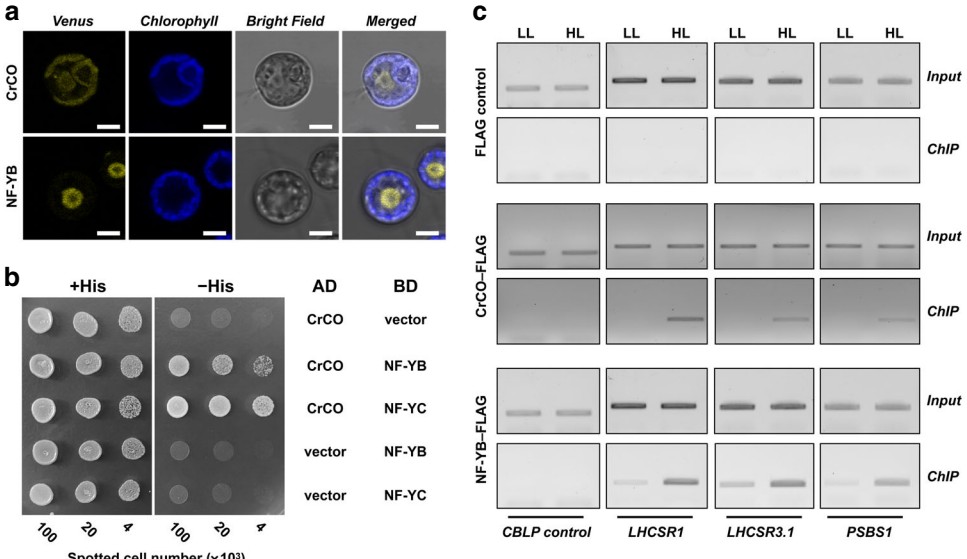

**Fig. 2** CrCO interacts with NF-Y isoforms to form a transcriptional complex that associates with the promoter regions of the photoprotective genes. **a** Confocal live-cell imaging of the complemented strains, crco-2/CrCO (labelled as CrCO) and nfyb-1/NFYB (labelled as NF-YB), to visualize the localization of CrCO and NF-YB fused with Venus–3xFLAG in *C. reinhardtii* cells. Scale bars, 5 μm. **b** GAL4-based yeast two-hybrid (Y2H) assays. CrCO was fused to the GAL4-activation domain (AD–CrCO), whose ability to form heterodimers was tested by cotransformation of the GAL4-binding domain fused with NF-YB or NF-YC (BD–NF-YB or BD–NF-YC). Autoactivation was tested using empty vectors (AD or BD). The cells were plated on permissive (+His) or selective (−His) media and grown at 30 °C for 40 h after being spotted on the plates. **c** Chromatin immunoprecipitation (ChIP)-PCR assay of CrCO and NF-YB. Agarose gel electrophoresis showing the strength of the association of CrCO or NF-YB with the promoter regions of the photoprotective genes under different light treatments. ChIP was performed using 2 × 10$^8$ cells/mL of crco-2/CrCO (labelled as CrCO) or nfyb-1/NFYB (labelled as NF-YB) cells cross-linked with 0.35% (v/v) formaldehyde after a 1-h light treatment

expression, whereas the nfyb-1/NFYB strain showed a normal photoprotective response (Fig. 1d, e). Considering that the expression kinetics of the key photoprotective proteins (LHCSRs and PSBS) are similar to CrCO protein accumulation (Fig. 1e), it is likely that the CrCO protein accumulation or localization in the crco-2/CrCO strain is altered. These results suggest that CrCO protein accumulation induced by light illumination was responsible for the photoprotective responses in *C. reinhardtii*.

**CONSTANS/NF-Y complex controls photoprotective transcription.** During the complementation analysis of crco-2 and nfyb-1, a Venus–3xFLAG tag was fused to the C termini of CrCO and NF-YB, which enabled intracellular live imaging of these proteins. Venus fluorescence was detected near the nucleus in the crco-2/CrCO and nfyb-1/NFYB strains. Immunocytochemistry analysis also showed that the DAPI and FLAG signals colocalized in the complemented strains (Supplementary Fig. 5), indicating that CrCO and NF-YB colocalized in the nucleus (Fig. 2a). Because complex formation among CO, NF-YB, and NF-YC has been reported to be essential for *FT* regulation in flowering plants[20], we further tested the possibility of the involvement of NF-YC in the complex by generating and characterizing the *NF-YC* (Cre12.g556400) mutants in *C. reinhardtii*. The CRISPR-Cas9-mediated mutation in *NF-YC* severely affected both the transcription and translation of the photoprotective factors, in a similar manner to the CrCO or NF-YB mutants. This resulted in cell death under HL (Supplementary Fig. 6). We subsequently tested the physical interactions between CrCO and NF-YB, CrCO and NF-YC, and NF-YB and NF-YC using yeast two-hybrid (Y2H) assays, suggesting that CrCO, NF-YB, and NF-YC interacted with each other (Fig. 2b and Supplementary Fig. 7). Similar tripartite interactions among the corresponding proteins have been reported in flowering plants[15,20,21]. These interactions were confirmed using immuno-coprecipitation assays of

CrCO–Venus–3xFLAG and NF-YB–Venus–3xFLAG with FLAG monoclonal antibody followed by LC-MS/MS spectroscopy. This revealed the presence of NF-YB and NF-YC in the precipitates of CrCO–Venus–3xFLAG and NF-YB–Venus–3xFLAG, respectively (Supplementary Tables 1 and 2). Moreover, CrCO was detected in the precipitate of NF-YB–Venus–3xFLAG (Supplementary Fig. 8). Together, these results strongly indicate that the CO/NF-YB/ NF-YC complex is conserved in *C. reinhardtii*.

Mutations in *CrCO* or *NF-YB/NF-YC* severely suppressed the transcription of the photoprotective genes in *C. reinhardtii* (Fig. 1 and Supplementary Fig. 6). Meanwhile, mutations in *CO* and/or *NF-YA/NF-YB/NF-YC* in flowering plants have been shown to reduce *FT* transcription[10,15,20]. All the photoprotective genes including *LHCSR1*, *LHCSR3.1/3.2*, and *PSBS1/2* have at least one CO-responsive element (CORE), *CCACA*[22], and NF-Y cis-element, *CCAAT*[12], in the upstream regions of their start codons (Supplementary Fig. 9). It is therefore plausible that CrCO and NF-YB act as transcription factors that regulate the expression of the photoprotective genes in *C. reinhardtii*.

To investigate whether CrCO and NF-YB associate with the promoter regions of the photoprotective genes (*LHCSR1*, *LHCSR3.1*, and *PSBS1*) in *C. reinhardtii*, we performed chromatin immunoprecipitation (ChIP) assays using the cells before and after HL treatment. Both CrCO and NF-YB associated with the promoter regions of these genes after HL treatment (Fig. 2c), which was quantitated using ChIP-qPCR analysis (Supplementary Fig. 10). NF-YB associated with the promoters of the photoprotective genes in LL, and the association seemed to be facilitated under HL (Supplementary Fig. 10). It has been suggested that the NF-YB/NF-YC complex in *A. thaliana* binds near CORE elements (*CCAAT* for NF-YA[12] and *CCACA* for CO[22]) to support the association of either NF-YA or CO with the promoter region[23]. Considering this binding feature of plant NF-YB/NF-YC, it is conceivable that NF-YB

weakly associates with the promoter region of the photoprotective genes under LL and that this association is reinforced by CrCO accumulation and the formation of a CrCO/NF-YB/NF-YC complex under HL. Together with the results of our physiological analysis (Fig. 1), this demonstrates that the CrCO/NF-YB/NF-YC transcriptional module is involved in photoprotective processes in *C. reinhardtii*.

**UVR8 inhibits E3 ubiquitin ligase to accumulate CONSTANS.** In land plants, CO protein accumulation is controlled by light-dependent post-transcriptional regulation[24]. This regulation is modulated by two CULLIN 4 (CUL4)-based E3 ubiquitin ligase components[25], SPA1 and COP1, which are involved in the ubiquitination-dependent degradation of CO[26,27]. The E3 ubiquitin ligase activity of this complex is deactivated by the formation of a protein–protein complex with photoactivated CRYPTOCHROME 2 (CRY2)[28]. The interaction between COP1 and the UV-photoreceptor UVR8 has also reported to be important for UV responses in both land plants[29] and *C. reinhardtii*[7]. Using a UVR8–Venus–3xFLAG expression strain, UV-inducible nuclear translocation of UVR8 was observed (Fig. 3a and Supplementary Figs. 5 and 11). Additionally, interactions among UVR8, SPA1, and COP1 were identified using a coimmunoprecipitation assay (Fig. 3b and Supplementary Tables 3 and 4). These results imply that upon UV perception, UVR8 deactivates E3 ubiquitin ligase through the formation of the UVR8/SPA1/COP1 protein complex as proposed in land plants[30].

As SPA1 and COP1 are involved in the ubiquitination-dependent proteasomal degradation of CO in land plants[26,27], we further investigated the possible interaction among SPA1, COP1, and CrCO. Both SPA1 and COP1 interacted with CrCO in Y2H assays (Fig. 3c). Additionally, UV-dependent accumulation of CrCO was observed in the *crco-2/CrCO* complemented strain, while it overexpressed *CrCO* mRNA even before UV illumination (Fig. 3d and Supplementary Fig. 12). It is therefore likely that CrCO is degraded in LL by proteasomes after ubiquitination by the SPA1/COP1-dependent E3 ubiquitin ligase complex. This degradation is then inhibited upon exposure to UV.

To further clarify whether the E3 ubiquitin ligase is involved in the accumulation of CrCO and photoprotective proteins, an *spa1* mutant was analyzed. This mutant has an insertional mutation in the *SPA1* gene, and was obtained from the *Chlamydomonas* mutant library project (*CLiP*) (see Methods for detail). This mutation mimics deactivation of the SPA1/COP1-dependent E3 ubiquitin ligase. As expected, overexpression of the photoprotective proteins (LHCSRs and PSBS) were observed even under LL conditions in this mutant owing to the accumulation of CrCO (Fig. 3d). Furthermore, an inhibition of proteasome activity by MG132 treatment of the *crco-2/CrCO* strain, which resulted in the accumulation of CrCO protein even under LL conditions (Supplementary Fig. 13), also led to the accumulation of several ubiquitin polypeptides (*Cre09.g396400*) identified by LC-MS/MS analysis of the coimmunoprecipitated samples of CrCO–Venus–3xFLAG (Supplementary Table 5). This indicates that CrCO is actively degraded by proteasomes that are guided by the ubiquitination of the protein. These data further support that SPA1/COP1 modules operate in distinct physiological functions, flowering time control in land plants and photoprotection in *C. reinhardtii*.

The CO/NF-YB/NF-YC complex plays a pivotal role in the photoperiodic control of flowering in land plants through transcriptional regulation of *FT* expression. Our findings

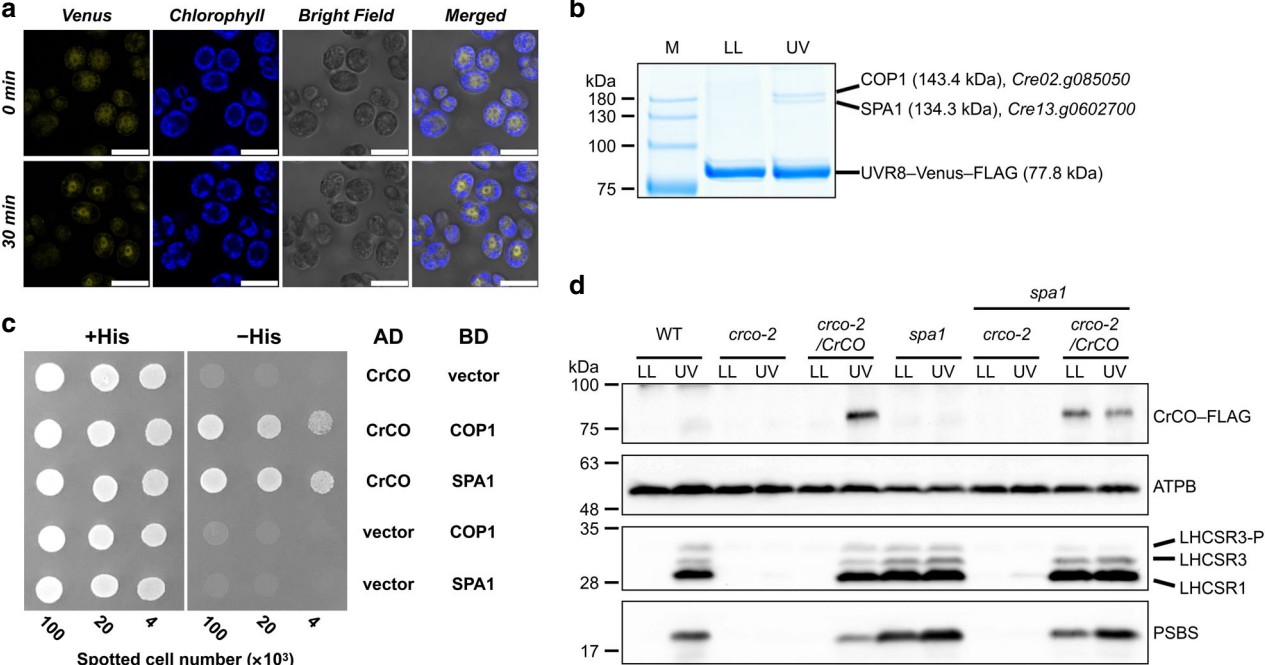

**Fig. 3** UVR8 interacts with COP1 and SPA1 to activate the CrCO-based transcriptional complex under UV irradiation. **a** Confocal images of UVR8–Venus–3xFLAG proteins in the *DSR1–comp15* (*uvr8/UVR8*) strain. The cells were treated with UV for 30 min. Scale bars, 10 μm. **b** After 1 h of UV treatment, the cells were harvested and UVR8–Venus–3xFLAG proteins were immunoprecipitated by FLAG (M2) antibody with SURE-beads. The coimmunoprecipitated proteins were identified by LC-MS/MS analysis of the Coomassie Brilliant Blue-stained polypeptide bands obtained via SDS-PAGE separation after in-gel trypsin digestion. M, molecular mass standard. **c** Interaction profiles among COP1, SPA1, and CrCO visualized with Y2H assays, which were performed using COP1, SPA1, and CrCO fused with the AD and/or BD domains of GAL4. The culture conditions were as described for Fig. 2b. **d** Immunoblot analysis of LHCSRs, PSBS, and 3xFLAG-fused CrCO to visualize protein accumulation in the wild-type control (WT), *crco-2*, *crco-2/CrCO*, *spa1*, *spa1 crco-2*, and *spa1 crco-2/CrCO* strains before and after UV treatment. ATPB proteins were included as the loading controls

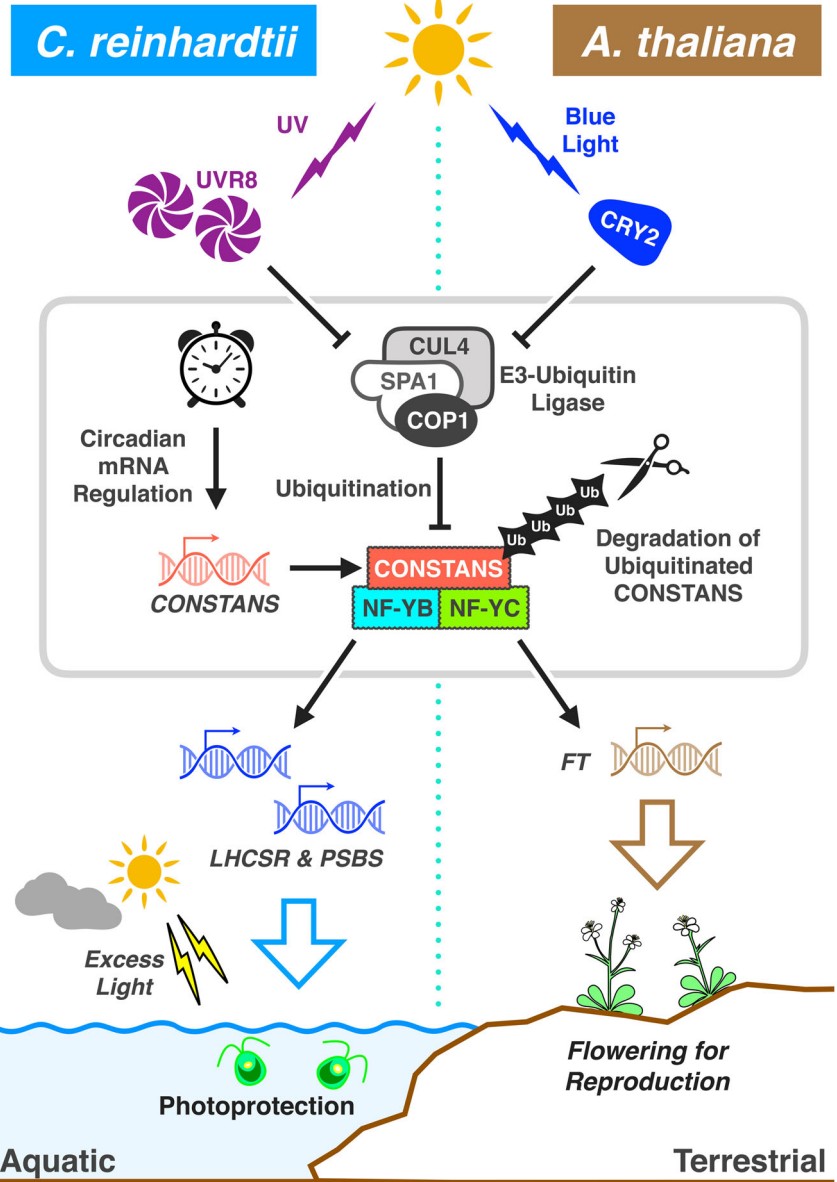

**Fig. 4** Hypothetical signal transduction pathways and physiological functions of CONSTANS/NF-YB/NF-YC-dependent control of the target genes in *C. reinhardtii* and *A. thaliana*. Blue light perception by the photoreceptor CRY2 in *A. thaliana* deactivates the E3 ubiquitin ligase module[27,28], whereas UV light perception via conversion of the dimeric form of UVR8 to the monomeric form deactivates this module in *C. reinhardtii*[7]. Circadian clock-regulated *CONSTANS* transcripts[10] are translated, and successfully accumulate in the organisms when proteasome- and ubiquitin-dependent degradation is inactivated. The CONSTANS/NF-YB/NF-YC transcriptional module then transcribes *FT*[10] and photoprotective genes (e.g., *LHCSR1*, *LHCSR3*, and *PSBS*) to stimulate flowering and photoprotection in land plants and the green alga, respectively. The kernel, including SPA1/COP1-dependent E3 ubiquitin ligase, circadian regulation of *CONSTANS*, and the CONSTANS/NF-YB/NF-YC transcriptional complex, was possibly established in an ancestral green photosynthetic organism

therefore suggest that there may also be a link between photoperiodic signaling and the photoprotective response in *C. reinhardtii*. The transcript levels of *CrCO* indeed increased during subjective daytime but decreased in subjective night time (Supplementary Fig. 14). This is in part compatible with the previous report that *CrCO* transcription is under the control of the circadian clock[16]. Previous studies have also revealed that *LHCSR3*, formerly known as *LI818*, displays diurnal oscillations in RNA accumulation under light/dark cycles[31,32]. As CrCO is critical for survival under HL (Fig. 1), CrCO in fact connects the circadian clock with photoprotective responses in *C. reinhardtii* (Fig. 4).

## Discussion

Aihara et al. recently reported that another E3 ubiquitin ligase complex, CUL4–DDB1[DET1], to be involved in the negative regulation of *LHCSRs* in *C. reinhardtii*[8]. Since a similar complex has been proposed to enhance the activity of the SPA1/COP1-dependent E3 ubiquitin ligase in land plants[25], the algal CUL4–DDB1[DET1] may also enhance the ubiquitination at the SPA1/COP1-dependent E3 ubiquitin ligase, thereby transducing the blue light signal to *LHCSRs* by way of CrCO. Taken together with the synchronized accumulation kinetics of CrCO and the photoprotective proteins in the *crco-2/CrCO* strain (Fig. 1e), we propose that the post-translational regulation of CrCO mediated

by the E3 ubiquitin ligase, is fundamental for the photoprotective response in *C. reinhardtii* (Fig. 4).

The initial light input signals differ between flowering plants (in which blue light is perceived by CRY2[28]) and *C. reinhardtii* (in which UV is perceived by UVR8). Despite this, the kernel of the downstream regulatory components (SPA1/COP1-dependent E3 ubiquitin ligase to CO/NF-Ys transcriptional complex) is shared (Fig. 4). Previously, the regulatory kernel has been proposed as part of a unique mechanism developed for photoperiodic flowering[11]. The results presented in this study have shown that the kernel's counterpart in *C. reinhardtii* functions in photoprotection. This raises the possibility that the root of this method of regulation could be extended back to the early evolutionary history of green photosynthetic organisms. At the same time, we may need to reinterpret the significance of flowering in plants and photoprotection in green algae within the same context, because of a shared fundamental regulatory mechanism. Whether we can see the same type of regulation in other algae would be a matter of future study as our current knowledge about their photoprotective behavior as well as the related genetic information is still very limited.

## Methods

**Statistics**. Statistical methods were not used to predetermine the sample size. The experiments were not randomized, and the investigators were not blinded to allocation during experimental procedures and data assessment.

**Strains and growth conditions**. *Chlamydomonas reinhardtii* strains 137c, *CC4286*, and *CLiP* (CC5325 as the wild type and *LMJ.RY0402.060340* as the *spa1* mutant) were obtained from the Chlamydomonas Resource Center (http://www. chlamycollection.org). All strains were grown in tris-acetate-phosphate (TAP) medium and illuminated at 50 μmol photons m$^{-2}$ s$^{-1}$ at 25 °C with rotary shaking at 150 rpm. Cells at mid-log phase (~$4 \times 10^6$ cells/mL) were harvested and resuspended in high salt (HS) medium at $1 \times 10^6$ cells/mL and treated with light as described in the text and figure legends. The *LHCSR1–Luc717* strain, harboring a LHCSR1–luciferase fused reporter construct[9], was used as a control strain for the *DSR* mutants. The 137c strain was used as a control/recipient strain for *NF-YC* mutants. All the *DSR* mutants were previously isolated through a forward genetic screen using the *LHCSR1–Luc717* strain[9].

**Growth evaluation of the strains**. The growth of WT and mutant strains used in this study was evaluated under the conditions previously described[16]. Briefly, cells were grown under 30 or 100 μmol photons m$^{-2}$ s$^{-1}$ of continuous white fluorescent light in HS media, and cells were counted every 24 h. To assess cell shapes, cells at the steady-state growth phase (day 6) were imaged at room temperature using a TCS SP8 confocal laser scanning microscope (Leica Microsystems, Germany) equipped with a HC PL APO CS2 63×/NA1.40 oil objective lens. A 448-nm diode laser with 2% output was selected to obtain bright-field images. The images were produced using a PMT-based detector. All images were acquired at a 600-Hz laser scan speed and analyzed by LASX software (Leica Microsystems).

**Identification of the insertion site of the *aph7* cassette**. The genomic regions flanking the inserted DNA tag were cloned by RESDA-PCR[33]. The tag DNA-specific and degenerate primers are listed in Supplementary Table 6. The PCR mixture for a first amplification contained tag-specific primer (*aph7tag-F1* for downstream of the tag DNA; *aph7tag-R1* for upstream of the tag DNA), degenerate primer (*DegPstI*, *DegSacII*, *DegBglII*, or *DegMluI*), rTaq DNA polymerase (TOYOBO, Japan), and genomic DNA template. The mixture was placed in a SimpliAmp Thermal Cycler (Thermo Fisher Scientific, Waltham, MA, USA), and the following program was run: 95 °C for 8 min; five cycles of 95 °C for 30 s, 58 °C for 1 min, and 72 °C for 3 min; 95 °C for 30 s; 25 °C for 3 min; 72 °C for 3 min; 20 cycles of 95 °C for 30 s, 58 °C for 1 min, 72 °C for 3 min, 95 °C for 30 s, 58 °C for 1 min, 72 °C for 3 min, 95 °C for 30 s, 40 °C for 1 min, and 72 °C for 3 min; 72 °C for 5 min. The PCR mixture for a second amplification contained tag-specific primer (*aph7tag-F2* or *aph7tag-F3* for downstream of the tag DNA; *aph7tag-R2* or *aph7tag-R3* for upstream of the tag DNA), Q0 primer, KOD FX Neo DNA polymerase (TOYOBO), and the diluted first amplification product. After running standard amplification program for KOD FX Neo DNA polymerase, the second amplification products were purified using the Wizard SV Gel and PCR Clean-Up System (Promega, Madison, WI, USA) and subjected to DNA sequencing. The sequenced genomic nucleotides were then analyzed by a BLAST search in *Chlamydomonas reinhardtii* genome v5.5 (Phytozome, https://phytozome.jgi.doe.gov/) to identify the position of the inserted *aph7* cassette in the genome.

**Linkage mapping of the *dsr4286* and *dsr28* mutations**. Having determined that the CC4286 strain harbored a mutation that caused a defect in UV-inducible *LHCSR1* expression[9], we named the mutation *dsr4286*. To establish the *dsr4286* mutation in the 137c background, CC4286 was crossed with *LHCSR1–Luc443*, another clone of *LHCSR1*–luciferase transformants of 137c (mt+). An F3 progeny, *#443-1A-12-4b* (mt + *dsr4286*), was crossed with the polymorphic strain S1D2 (mt−, CC2290[34]), and tetrad progeny were dissected. Recombination frequencies between *dsr4286* and genetic markers were determined by detecting polymorphic PCR products in 44 progeny that were deficient in UV-inducible *LHCSR1* expression (*dsr* phenotype). *dsr4286* was mapped to a region before position 1515000 on chromosome 2 of the Joint Genome Institute (JGI) version 5.5. To examine the linkage between the *dsr4286* and *dsr28* mutations, *#443-1A-12-4b* was crossed with an F1 progeny of a backcross of *DSR28* and *LHCSR1–Luc717* (mt−). All the resulting 91 progeny were tested for *LHCSR1* expression and showed the mutant phenotype, suggesting that mutation in *DSR28* (*dsr28*) is strongly linked to *dsr4286*.

**Whole-genome sequencing of *dsr4286* and *dsr28***. Two *dsr4286* mutants, the F3 progeny of a backcross of CC4286 [*#443-1A-12-5a* (*dsr4286* mt+) and *#443-1A-12-5d* (*dsr4286* mt−)], and four clones from one tetrad set of the second backcross of *DSR28* [*DSR28-7c-4a* (*dsr28* mt−), −4b (wild-type mt+), −4c (wild-type mt−), and −4d (*dsr28* mt+)] were subjected to whole-genome sequencing analysis. The genomic DNA was prepared using a DNeasy Plant Mini Kit (Qiagen, Germany). Libraries were constructed using an Illumina TruSeq DNA PCR-Free Library Prep Kit (Illumina, San Diego, CA, USA) from 2 μg of each DNA sample, according to the low-throughput protocol. Sequencing was conducted on a HiSeq 1500 sequencer (Illumina) to produce $2 \times 106$-bp paired-end reads. The sequencing data were submitted to the DDBJ Sequence Read Archive (SRA) (http://www.ddbj. nig.ac.jp/dra). All sequences were trimmed with Trimmomatic[35] and aligned onto the JGI version 5.5 *Chlamydomonas* assembly using Bowtie2 (bowtie-bio.source-forge.net/bowtie2/index.shtml). The resulting files were converted to Binary Sequence Alignment/Map and sorted by SAMtools (samtools.courveforge.net/). The variants were identified using SAMtools and freebayes (https://github.com/ekg/freebayes/).

To identify the variant responsible for the *dsr4286* mutation, the variants that were mapped to a region before position 1515000 on chromosome 2 were extracted from *#443-1A-12-5a* (*dsr4286* mt+), *#443-1A-12-5d* (*dsr4286* mt−), *dsr28-7c-4b* (wild-type mt+), and *dsr28-7c-4c* (wild-type mt−) strains for comparison. Twenty variants were identified that were present in *#443-1A-12-5a* and *#443-1A-12-5d* but not in *DSR28-7c-4b* (wild-type mt+) or *DSR28-7c-4c* (wild-type mt−). These variants were confirmed on the alignment data visualized by the Integrated Genomics Viewer (IGV) (http://software.broadinstitute.org/software/igv/). The variant at position 871101 located in the first exon of *Cre02.g079200* appears to be a causal mutation of *dsr4286* because all 31 reads contained a 5-bp deletion. To identify a variant responsible for the *dsr28* mutation, the alignment data of *DSR28-7c-4a* (*dsr28* mt−) and *DSR28-7c-4d* (*dsr28* mt+) were visualized by IGV, with a focus on *Cre02.g079200*. The ~30-bp region around position 871830 on chromosome 2, located in the third intron of *Cre02.g079200*, was not mapped in either strain.

To further confirm the mutation sites of *dsr4286* and *dsr28*, PCR amplification followed by sequencing analysis was performed. For the *dsr4286* mutation, ~1-kb DNA fragments were PCR amplified using *DSR28-EcoRI-F* and *DSR28-R2* primers, and genomic DNA was prepared from strains *#443-1A-12-5a* and *#443-1A-12-5d* as a template. Sequencing analysis of these fragments revealed that 5 nt was missing from the first exon (Supplementary Fig. 2). For the *dsr28* mutation, primers *Cre12 (2898864R)* and *Cre12(2898864R)* were designed to anneal the insertion predicted by the visualized mapping on IGV. The ~700- and ~500-bp fragments were PCR amplified using primer pairs *DSR28-F2* and *Cre12(2898864R)*, and *Cre12 (2894770F)* and *DSR28-R2*, respectively, and genomic DNA was prepared from *DSR28-7c-4a* and *DSR28-7c-4d* as a template. Sequencing analysis revealed that the third intron of *Cre02.g079200* contained the 33-bp deletion (Supplementary Fig. 2).

**NFYC gene disruption using the CRISPR-Cas9 system**. CRISPR-Cas9 targeting of *NFYC* exon2 was performed to generate *nfyc* mutants as described previously[36], using a guide RNA sequence (5′-TGAATACAGTTGGGTCAGGT-3′) and a double-stranded homology-directed repair donor (ds-HDR). The sequences of sense (HDR-F) and antisense (HDR-R) 90-bp oligonucleotides were arranged with three PTO bonds at the 3′ and 5′ end bases. Equimolar concentrations of sense and antisense oligonucleotides were annealed in 1× DUPLEX buffer (100 mM potassium acetate, 30 mM HEPES, pH 7.5, Integrated DNA Technologies [IDT; Coralville, IA, USA]) by incubation at 95 °C for 2 min to generate 10 μM ds-HDR. Recombinant *Streptococcus pyogenes* Cas9 protein, tracrRNA, and crRNA were purchased from IDT. Equimolar concentrations of tracrRNA and crRNA were annealed in DUPLEX buffer by heating to 95 °C for 2 min to generate 10 μM gRNA. Subsequently, 10 μM Cas9 protein was mixed with equimolar concentrations of annealed gRNA in 1x Buffer 3.1 (New England Biolabs, Beverly, MA, USA) to a final concentration of 3 μM each and incubated for 15 min at 37 °C to form ribonucleoprotein (RNP). Cells were grown in a synchronized light cycle (light for 14 h at 25 °C: darkness for 10 h at 18 °C) in TAP medium and then harvested and suspended in MAX Efficiency Transformation Reagent for Algae (Thermo Fisher Scientific) supplemented with 40 mM sucrose to a density of $1 \times 10^8$ cells/mL. The

concentrated cells were heat-shocked at 40 °C for 30 min and mixed using a thermoshaker (350 rpm). Forty microliters of heat-shocked cell suspension was mixed with 5 μL of 3 μM RNP, 1.5 μL of 10 μM ds-HDR, and 0.3 μg of selection marker plasmid pHyg3 and then electroporated using the NEPA21 Super Electroporator (Nepa Gene Co., Ltd., Japan). The transformed cells were then selected under dim light at 25 °C on TAP plates containing 10 μg/mL of hygromicin. The hygromycin-resistant clones were screened for insertion of the "FLAG" sequence by colony PCR using *NFYC*-check-F and *FLAG*-R primers.

**Complementation of *DSR* mutants**. For the complementation of *crco-2* (*DSR15*), *nfyb-1* (*DSR28*), and *uvr8* (*DSR1*) mutants, a full-length version of their genes was fused in-frame to Venus–3xFLAG under the control of the *HSP70A/RBCS2* tandem promoter[37]. The generated complementation vectors with the *aadA* gene (spectinomycin resistance marker) were introduced into the *DSR15*, *DSR28*, or *DSR1* strains by electroporation using a NEPA21 Super Electroporator (Nepa Gene Co., Ltd.). All strains in the exponential phase of growth ($2 \times 10^6$ cells/mL) were transformed with 300 ng of linearized vector. The transformed cells were then selected under dim light at 25 °C on TAP plates containing 10 μg/mL of spectinomycin.

**Genetic crossing of the mutants**. To generate double mutants harboring the *spa1* mutation and *CrCO–Venus–3xFLAG* construct, the *CLiP spa1* mutant *LMJ. RY0402.060340* was crossed with the *crco-2/CrCO* strain. The resulting progeny were dissected and categorized to wild-type, *spa1*, *crco-2/CrCO*, *spa1 crco-2*, and *spa1 crco-2/CrCO* genotype strains. To generate triple mutants harboring the *crco-2* mutation, *nfyb-1* mutation, and *NFYB–Venus–3xFLAG* construct, the *crco-2* mutant was crossed with the *nfyb-1/NFYB* strain. The resulting progeny were dissected and *crco-2 nfyb-1/NFYB* genotype strains were isolated.

**Standard mRNA quantification**. Total RNA from light-treated cells was extracted using the Maxwell RSC instrument (Promega) equipped with the Maxwell RSC simplyRNA Tissue Kit (Promega). The isolated RNA was quantified using the QuantiFluor RNA System (Promega) prior to reverse transcription. Reverse transcription and semiquantitative PCR were performed using a ReverTra Ace® qPCR RT Kit with gRemover (TOYOBO) and KOD FX Neo DNA Polymerase (TOYOBO) in the SimpliAmp Thermal Cycler (Thermo Fisher Scientific). For reverse transcription, a 300 ng of isolated RNA sample was used in a 10-μL total reaction volume. Semiquantitative RT-PCR was performed using 3 ng of cDNA thus obtained. For semiquantitative RT-PCR, the G protein β-subunit-like polypeptide (*CBLP*) gene was chosen as the housekeeping gene during light treatment. The primers used for semiquantitative RT-PCR are listed in Supplementary Table 7.

**Immunoblot analysis**. Protein samples from whole cell extracts (corresponding to ~$2 \times 10^6$ cells, unless stated otherwise) were loaded onto 11% acrylamide with 7 M urea SDS-PAGE (gel size: 15 cm width × 10 cm height × 1 mm thickness), and the electrophoresis was performed at a constant current 8 mA for 15 h. The separated polypeptides were blotted onto nitrocellulose membranes with a tank blot system (BIO-CRAFT, Japan) at a constant voltage 40 V for 3 h. The blotted membranes were treated with a blocking reagent EzChemi Block (ATTO, Japan) for 1 h. The primary and secondary antibody treatments were done with shaking at room temperature for 1 h. Antiserum against ATPB was obtained from Agrisera (AS05 085, rabbit polyclonal, at 1:10,000 dilution); antisera against LHCSR1 and LHCSR3 (rabbit polyclonal, at 1:10,000 dilution) were raised and affinity purified against the peptide LGLKPTDPEELK;[9] antiserum against PSBS (rabbit polyclonal, at 1:5000 dilution) was provided Prof. Peter Jahns (Heinrich-Heine-University, Germany); antisera against CrCO (rabbit polyclonal, at 1:1000 dilution) was raised and affinity purified against the peptide AAWFVDDEKMG (Eurofins Genomics, Japan); and antiserum against FLAG fusion proteins was obtained from Sigma-Aldrich (F1804, mouse monoclonal, at 1:1000 dilution). An anti-rabbit horseradish peroxidase-conjugated antiserum (#7074, Cell Signaling Technology, Danvers, MA, USA) or an anti-mouse horseradish peroxidase-conjugated antiserum (#330, MBL, Japan) was used as a secondary antibody (at 1:10,000 dilution). The blots were developed using ECL detection reagent EzWestLumi Plus (ATTO), and images of the blots were obtained using a CCD imager ChemiDocTouch System (Bio-Rad Laboratories, Hercules, CA, USA). The upper band of LHCSR3 represents its phosphorylated form (LHCSR3-P)[6].

**High-light tolerance assay**. Mutant cells suspended in HS medium were irradiated for 8 h with either low light (LL; white fluorescence light at 30 μmol photons m$^{-2}$ s$^{-1}$) or high light (HL) at 300 μmol photons m$^{-2}$ s$^{-1}$ including UV (using a T8 ReptiSun 10.0 UVB fluorescent bulb [Zoo Med Laboratories, CA, USA] at 30 μmol photons m$^{-2}$ s$^{-1}$) at 25 °C. Total cell protein extracts were obtained from the cell cultures after 1, 2, 4, 6, and 8 h of light treatment. Chlorophyll concentrations and cell numbers were calculated by the Porra calculation[38] and TC20 automated cell counter (Bio-Rad Laboratories). The chlorophyll levels were then normalized to cell numbers (pg of chlorophyll per cell). LL- and HL-treated strains were adjusted to $1 \times 10^7$ cells/mL, transferred into multiwell plates, and photographed, as indicated in the figure captions (Fig. 1 and Supplementary Fig. 6).

**Chlorophyll fluorescence-based photosynthetic analysis**. The cells were transferred into a 48-muliwell plate and treated with far-red light (<5 μmol photons m$^{-2}$ s$^{-1}$) for 30 min. To evaluate the maximum quantum yield of photosystem II (Fv/Fm), minimal and maximal fluorescence yields (Fo; Fm) were measured under dark conditions using a FluoCam MF800 (Photon System Instruments, Czech Republic). Fv/Fm was calculated using the equation Fv/Fm = (Fm − Fo)/Fm. To evaluate the photoprotection capability (qE), maximal fluorescence yield under light (Fm′) was also measured after actinic irradiation at 750 μmol photons m$^{-2}$ s$^{-1}$ for 30 s. qE was estimated using the equation qE = (Fm − Fm′)/Fm′.

**Yeast two-hybrid assay**. To construct the GAL4-activation domain (AD)- or GAL4-binding domain (BD)-fused *CrCO*, *NF-YB*, *NF-YC*, *SPA1*, or *COP1* expression plasmids for the yeast two-hybrid assay, the corresponding full-length coding sequences were amplified from cDNA obtained from *Chlamydomonas* strain 137c by PCR (primers CrCO-EcoRI-F/CrCO-SalI-R for CrCO, NF-YB-EcoRI-F/NF-YB-SalI-R, NF-YC-EcoRI-F/NF-YC-SalI-R, COP1-F-Y2H/COP1-R-Y2H for COP1 and SPA1-F-Y2H/SPA1-R-Y2H for SPA1) using KOD FX Neo DNA polymerase (TOYOBO) on a SimpliAmp Thermal Cycler (Thermo Fisher Scientific). The amplicons (~1.25 kb for *CrCO*, ~0.68 kb for *NFYB*, ~0.91 kb for *NFYC*, ~4.37 kb for *COP1* and ~4.17 kb for *SPA1*) were cloned into the pGAD-C2 or pGBD-C2 vectors[39]. AD and BD fusion plasmids were introduced into *Saccharomyces cerevisiae* reporter strain pJ69-4A (*MAT*a *trp1 leu2 ura3 his3 gal4Δ gal80Δ LYS2::GAL1-HIS3 GAL2-ADE2 met2::GAL7-lacZ*)[39]. The transformants were then selected on synthetic dextrose (SD) plates lacking tryptophan (Trp) and leucine (Leu) and used for the yeast two-hybrid assay. An empty AD or BD vector served as a negative control. The yeast cells were suspended in liquid medium at a concentration of $2 \times 10^7$ cells/mL, and 5-μL droplets of 5-fold serial dilutions were spotted onto SD-Leu-Trp (+His) and SD-Leu-Trp-His (−His) plates. The plates were incubated at 30 °C for 40 h. The primers used for PCR analyses are listed in Supplementary Table 8.

**Immunoprecipitation and mass spectroscopy analysis**. Cells treated with each experimental light condition (HL for *nfyb-1/NFYB* and UV for *uvr8/UVR8* and *crco-2/CrCO*) for 1 h were harvested and resuspended in standard PBS buffer. The cells were disrupted by a previously reported airbrush method[40] using a 0.2-mm-caliber airbrush (PS270, GSI Creos Corporation, Japan) three times at 7.5 kgf cm$^{-2}$ and then solubilized with Triton X-100 (Sigma-Aldrich, St. Louis, MO, USA) for 5 min to a final concentration of 0.5% in PBS buffer. Unsolubilized cell debris was removed by centrifugation at 20,000 × g at 4 °C for 5 min. The Venus–3xFLAG-fused proteins were immunoprecipitated using 100 μL of SURE-beads (Bio-Rad Laboratories) conjugated with 5 μg of FLAG (M2) mouse monoclonal antibody (Sigma-Aldrich) at 4 °C for 1 h. The immunoprecipitates were then eluted using 3xFLAG peptide (ProteinArk, Sheffield, UK) at 500 ng/μL with shaking at room temperature for 15 min, and the eluted polypeptides were separated on a 12.5% PAGEL-HR gel (ATTO) at a constant current 10 mA for 90 min and in-gel trypsin digested according to Shevchenko et al.[41]. The in-gel-digested samples were subjected to LC-MS/MS spectroscopy analysis using an Orbitrap Elite mass spectrometer (Thermo Fisher Scientific). Tandem mass spectrometry results were submitted for protein database searching using Proteome Discoverer software (Thermo Fisher Scientific) and Mascot v.2.5.1 (Matrix Science, London, UK) against a database generated from *Chlamydomonas reinhardtii* genome v5.5 (Phytozome, https://phytozome.jgi.doe.gov/).

**Chromatin immunoprecipitation PCR assay**. Chromatin immunoprecipitation (ChIP) experiments were performed as reported previously[42]. Cells treated for 1 h in the high-light tolerance assay were harvested by centrifugation at 2000 × g at 4 °C for 2 min and resuspended in KH buffer (20 mM HEPES-KOH, pH7.6, and 80 mM KCl) containing 0.35% formaldehyde (Wako Chemical, Japan). After incubation for 10 min at room temperature, the cross-linking reaction was terminated with 1 M glycine for 5 min. The cross-linked cells were sonicated using the BIORUPTOR® II (Cosmo Bio, Japan) for 10 s (×30) with pauses of 20 s between each sonication cycle and then subjected to ChIP. The target DNA–protein complexes were immunoprecipitated using 100 μL of SURE-beads (Bio-Rad Laboratories) conjugated with 5 μg of FLAG (M2) mouse monoclonal antibody (Sigma-Aldrich) at 4 °C for 1 h. After an overnight cross-link reversion between DNA and protein at 65 °C, the DNA was extracted and purified using the phenol/chloroform/isoamylalcohol purification method. The purified DNA was then subjected to semiquantitative PCR using KOD FX Neo DNA polymerase (TOYOBO) on the SimpliAmp Thermal Cycler (Thermo Fisher Scientific). Real-time qPCR assays were performed using the KOD SYBR® qPCR Mix (TOYOBO) on the Light Cycler 96 system (Roche Diagnostics, Germany). The primers used for PCR analyses are listed in Supplementary Table 7.

**Live-cell imaging of Venus–3xFLAG fusion proteins**. *Chlamydomonas reinhardtii* cells expressing Venus–3xFLAG were mixed with low-gelling temperature agarose (A6560, Sigma-Aldrich) and transferred into a collagen-coated grass bottom dish (D11134H, Matsunami Glass, Japan). The agarose-embedded cells were imaged at room temperature on a TCS SP8 confocal laser scanning microscope (Leica Microsystems) equipped with a HC PL APO CS2 63×/NA1.40 oil objective lens. A 514-nm diode laser with 2% output was used to excite both Venus and chlorophyll,

and emission from 516 to 565 nm and 680–700 nm was collected for Venus fluorescence and chlorophyll autofluorescence, respectively, on a HyD SMD hybrid detector (Leica Microsystems). Bright-field images were produced using a PMT-based detector. All images were acquired with four-line accumulation at a 600-Hz laser scan speed in photon counting mode and analyzed using LASX software (Leica Microsystems). Cells expressing UVR8–Venus–3xFLAG in the *DSR1* (*uvr8*) background were imaged after 0, 5, 10, 15, 20, and 30 min of UV illumination.

**Immunocytochemistry of Venus–3xFLAG fusion proteins.** Cells were harvested by centrifugation at $2000 \times g$ at 4 °C for 2 min. and fixed by suspension in PBS containing 1% paraformaldehyde at 4 °C. The fixed cells were seeded on poly-L-lysine-coated coverslips (Asahi Techno Glass Corp., Japan) and were treated with methanol twice each for 10 min at −20 °C to permeabilize the membrane and remove chlorophyll. After rehydration with PBS, the coverslips were treated with a blocking buffer (5% bovine serum albumin, 1% cold water fish skin gelatin, and 10% goat serum in PBS), a primary antibody (mouse monoclonal anti-FLAG (M2) antibody; Sigma-Aldrich) at 4 °C overnight, and a secondary antibody [Alexa-Fluor546-conjugated F(ab')2 fragment of goat anti-mouse IgG; Thermo Fisher Scientific] at room temperature for 2 h. Coverslips were then mounted onto the slides with ProLong Diamond Antifade Mountant with DAPI (Thermo Fisher Scientific). Fluorescence was observed with a confocal laser scanning fluorescence microscope (FV10i-DOC, Olympus, Japan).

**Proteasome inhibitor assay.** The proteasome inhibitor MG132 (Wako Chemical) dissolved in dimethylsulfoxide (DMSO) was added to the culture before starting the light treatment. The final concentration of DMSO was adjusted to 0.5% with a final concentration of MG132 of 0, 10, 25, 50, 100, and 200 μM, as indicated in Supplementary Fig. 13. The cells were cultured under LL for 2 h after MG132 was added.

**Circadian mRNA rhythm assay of CrCO.** Asynchronous cultures in HS media ($2 \times 10^6$ cells/mL) were kept in darkness for 12 h at 17 °C to synchronize the circadian clock and then exposed to continuous light (10 μmol photons $m^{-2} s^{-1}$). Cells were harvested every 4 h from 24 to 72 h. mRNA quantification by RT-qPCR was performed as described previously[14]. Total RNA was extracted using TRIzol® reagent (Thermo Fisher Scientific). ReverTra Ace® qPCR RT Master Mix (TOYOBO) was used for the reverse transcription procedures. qPCR was then performed using KOD FX Neo DNA polymerase (TOYOBO) supplemented with SYBR® Green I Nucleic Acid Gel Stain (TaKaRa, Japan) and ROX dye (Thermo Fisher Scientific) to final concentrations of $0.05 \times$ (200,000-fold dilution) and 0.5 μM, respectively. Samples were quantified using StepOnePlus™ (Thermo Fisher Scientific) by the relative standard curve method. Quantification standards for each target cDNA were obtained using band-purified PCR products as templates. The constitutively expressed *RCK1* gene was used as an endogenous control for normalization. The primers used for PCR are listed in Supplementary Table 7.

**Phylogenetic analysis of NF-Y proteins.** The whole amino acid sequences of the NF-Y proteins from *C. reinhardtii* and *A. thaliana* were aligned using MAFFT v7.407 (https://mafft.cbrc.jp/alignment/software/). The phylogenetic tree was generated with RAxML v8.2.12 (https://cme.h-its.org/exelixis/web/software/raxml/index.html) with a 1000-bootstrap test. The generated phylogenetic tree was constructed and visualized with MEGA7 (https://www.megasoftware.net/).

**Reporting summary.** Further information on research design is available in the Nature Research Reporting Summary linked to this article.

## Data availability

The sequencing data have been deposited in [DDBJ Sequence Read Archive (SRA) with the accession codes DRX140139 (*#443-1A-12-5a*), DRX140140 (*#443-1A-12-5d*), DRX140141 (*DSR28-7c-4a*), DRX140142 (*DSR28-7c-4b*), DRX140143 (*DSR28-7c-4c*), and DRX140144 (*DSR28-7c-4d*)]. The proteomic data that support the findings of this study are available in [Japan ProteOme STandard Repository (jPOSTrepo) with the accession codes JPST000646 (Supplementary Table 1), JPST000647 (Supplementary Table 2), JPST000648 (Supplementary Table 3), JPST000649 (Supplementary Table 4), and JPST000650 (Supplementary Table 5). The source data underlying Figs. 1b–e, 2c, 3b, 3d and Supplementary Figs. 2b, 3, 4a, 6c, 6d, 9, 10, 12, 13, and 14 are provided as a Source Data file.

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

## Acknowledgements
We are grateful to Drs. Shuji Shigenobu, Shinichiro Maruyama, and Katsushi Yamaguchi for their help with whole-genome sequence analysis of the NF-YB mutants. We also appreciate Mrs. Yuko Mori and Ms. Yumiko Makino for their technical support with LC-MS/MS analysis. We also thank Mrs. Tamaka Kadowaki and Mrs. Harumi Yonezawa for providing technical assistance with the genetic crossing and culturing of the algae. We thank Dr. Masafumi Hirono for the technical advice on the genetic mapping and linkage analysis. We thank Ms. Ayumi Kinoshita for her help with the rhythm assay. The PSBS antibody was kindly provided by Dr. Peter Jahns. Drs. Kyoko Hamada, Kensuke Kataoka, and Junichi Nakayama are thanked for the fruitful discussions about the ChIP assay. We thank Dr. Krishna Niyogi for sharing results prior to publication. We also thank Drs. Kenji Takizawa, Yosuke Tamada, Tomonao Matsushita, and Ryoichi Sato for the fruitful discussion and critical reading of the manuscript. This work was supported by Functional Genomics Facility, NIBB Core Research Facilities, NIBB Model Plant Research Facility, NINS Program for Cross-disciplinary Study (grant number 01311701 to R.T.), and the JSPS Grant-in-Aid for Young Scientists (A) (grant number JP15H05599 to R.T.) and for Scientific Research on Innovative Areas (grant number JP16H06553 to J.M. and R.T.).

## Author contributions
R.T. and J.M. conceived the research. R.T. designed the experiments. R.T., K.F-K. and T.Y. screened and identified the mutation sites of the mutants. R.T. performed the transcriptional, ChIP-PCR, chlorophyll fluorescence, pigment bleaching, live-cell imaging, biochemical, and phylogenetical analyses. K.F-K. performed the algal linkage mapping, whole-genome sequencing, yeast two-hybrid analyses, and mutant generation. T.M. performed the immunocytochemistry and analyzed the circadian clock response of the green alga. R.T. wrote the initial draft of the paper. J.M. supervised the entire work and revised the paper. All authors contributed to the writing and revision of the paper and approved the final version of the paper.

## Additional information

**Competing interests:** The authors declare no competing interests.

