## [Peer Review File · Nature Communications]

Reviewers' comments:

Reviewer #1 (Remarks to the Author):

Review to Manuscript#: NCOMMS-19-01105-T by R Tokutsu et al 'The CONSTANS flowering module in algal photosynthesis'

1. Comments for Authors

Tokutsu and colleagues describe the identification of mutants affected in high light response in *Chlamydomonas reinhardtii* and the involvement and conservation of the plant CONSTANS homologue CrCO and plant NF-YB and NF-YC homologues in photoprotection. Employing biochemical and molecular biology tools they also identify a possible role for UV receptor UVR8 and E3 ubiquitin ligases homologues of COP1 and SPA1 in this response. They claim then that a conserved photoprotection signal involving this regulatory protein interaction network is conserved between algae and plants.

The work is very interesting and adds important information to the field. However, the actual version raises serious concerns that need to be addressed in detail. For clarity reasons I will separate major from minor points.

Major points.

- The conclusion made by the authors that CrCO and CrNF-YB mutants affect only photoprotection genes is not completely correct. It has been described that CrCO mutation and overexpression affect cell division and starch metabolism in algae and indeed a severe CrCO reduction or overexpression caused extreme growth phenotypes in the recombinant *Chlamydomonas* strains in high light. Previous authors had exposed the algae to different photoperiods in minimal media and the lethal phenotype was explained as a severe lack of photoperiod response leading the algae to incorrect growth, division, reproduction and eventually death. If photoprotection is one more of these aspects the others cannot be ignored. In order to put the data into a general context, I suggest a more massive expression analysis should be carried out. Microarrays or RNAseq experiments between wild type and mutant strains (as well as overexpressor lines) should be carried out and the connection with photoprotection deduced from the total.
- The interactions between CrCO and NYs, CrCO and CrCOP1 need a deeper characterization that Yeast-two-hybrid data. I suggest co-localization, BiFC or FRET data in order to quantify the interactions. Co-IP in which the actual presence of the protein is demonstrated only for NF-YC-B with mass data is not enough as this is a very sensitive and not quantifiable process. CO-IP followed by Immunodetection is an effective and measurable method as well. No data other than Y2H is shown for CrCO-NFs and CrCO-CrCOP1/SPA1 interaction.
- Lines 123-128. I think this is a major assumption. The paper or the literature does not show any data that this is true and this cannot be supported by a model as suggested in line 125 '...appear to be conserved (Fig. 4), ...'. Even when I tend to agree with the authors that this might be true, still more consistent data should be shown to support this.
- Complementation of CrCO and NF-YB mutants is achieved with the corresponding genes expressed under the control of the dual promoter RBs/Hsp70. This promoter is affected by regular light (and probably UV light) as can actually be seen in the amount of transcripts in supplemental data (Fig S7a) induced by this promoter. No claim of continuous expression without the correct controls can be assumed by the authors. Use of a UBQ or other constitutive promoters (or better its own promoters) should be tested.
- Lines 135-136. While CO expression is controlled by the clock through GI/FKF1 action in *Arabidopsis*, in *Chlamydomonas* a clock-control of CrCO expression is suggested by its reported photoperiodic-constant light expression analysis reported previously only. However, no effect of CO or CrCO in the clock has been described so far. This should be corrected.
- Materials and methods section and supplemental data FigS4 mention the edition of NF-YC gene and the phenotype resulting therein. Considering the importance of this experiment it is remarkable that no mention is made in the text. Authors should describe the figure, even consider adding as a main text, as well as describe and discuss these results in the text.

- In the confocal images showing the nuclear localization of proteins no control of the nuclei exists. Authors should use a dye or a known control to demonstrate that the actual distribution of CrCO, CNYs and CrCOP1/SPA1 are indeed imported into the nucleus. This is particularly important for the nuclear import of COP1 to the nucleus shown in Fig. 3a and S6.
- The rhythmic expression of CrCO has been already demonstrated in previous literature even in LL conditions, authors should clarify why the figure S8 is needed at all.

Minor points.

- The title needs to be modified. The flowering module concerning CONSTANS, involve the actual CDF-CO-FT expression control, so that a hierarchical order seemed to be needed to detect and regulate the photoperiodic flowering response. This module is partially conserved in algae, where a homologous DOF transcription factor similar to CDFs (CrDOF) control CrCO expression. However, the relationship described in this work cannot be considered a module because this is a protein complex and no hierarchy or "module" has been described. I would suggest a change in the title to something like 'The CONSTANS flowering complex in algal photosynthesis' or similar meaning.
- line 49, '..., these findings imply that the...'. This implication is not clear from the data presented in this work. I would rather use the term 'suggest'.
- In page 4, first lane, authors refer to a Fig 5, but this does not exist in the figure legends or in any actual picture sent with the manuscript. Please correct or delete.
- Fig 4: in the model presented by the authors it is implicated that CrCO is ubiquitinated and degraded by the proteasome. This is only deduced by the interaction with CrCOP1 but there is no actual data to confirm this point, so it cannot be described as such in the model. It is also assumed that CR2 affect CrCO protein through CrCOP1/CrSPA1 but this is also an assumption as no such data has been still published to my knowledge in Chlamydomonas.
- In materials section, immunoblot description; protein concentration is measured by assuming 1 microgram of chlorophyll. While this could be a poorly assumable protein concentration measure for the same cells in the same repetitive conditions, this cannot be the standard for different cells with different recombinant elements and particularly when dealing with light-dependent processes where chlorophyll and protein content may differ enormously. Proteins should be measured by standard methods and described in each individual blot and figure.
- Immunoprecipitation experiments are not described in sufficient detail. i.e. 'Cells treated with light' means standard light, UV light, low light, blue light? Buffer for protein solubilization (other than in the presence of triton X-100) is not described.
- For protein gels, corresponding size and molecular mass standards should be shown in the pictures. Sometimes more than 1 band are shown and this is not clarified in the text.
- CrCOP1 and CrSPA1 cloning is not fully described in the text. Size of the amplified fragments, the way they were isolated (or borrowed from another laboratory) should be described or acknowledged in the text.
- Negative controls for CrCO or CrNF-YB ChIP experiments are not described. A promoter where the binding of CO/NF complex is not light dependent or a promoter where the complex does no bind should be tested as well.
- Supplemental data is referred in some sentences as 'extended data Fig. n', while in others, the same picture is called 'Fig. Sn', please correct.
- mRNA methods. How much RNA was used to make cDNA to test for gene expression? Was it always the same quantity?

Reviewer #2 (Remarks to the Author):

The manuscript "The CONSTANS flowering module in algal photosynthesis" by Tokutsu, Fujimura-Kamada, Matsuo, Yamasaki, and Minagawa deals with the evolution of the transcriptional complex

that regulates flowering in land plants.

A large number of experiments has been performed, the results are in general robust and clearly presented.

1. I have one major concern with the general significance of the work. qE response in *Chlamydomonas reinhardtii* is induced when cells are exposed to strong illumination that induce the accumulation of LHCSR/PSBS. This is not true for most plants and eukaryotic algae where qE is constitutively present.

Now that more data on different species are available it is clear that *Chlamydomonas* has a peculiar behavior in this respect.

There is no evidence to my knowledge that signaling pathway inducing qE is conserved in other photosynthetic organisms. It is thus possible that results presented here are only valid for this species, questioning the general significance of the work.

2. In all manuscript data obtained with *Chlamydomonas* are referred as valid for all green algae. see as example lines 70-71: "In green algae, qE-type photoprotection is a light-inducible process that involves the expression of photoprotective proteins^{19,20}.". in particular for qE and regulation of photosynthesis we know since a few years that *Chlamydomonas* has peculiar behavior (as example see the xanthophyll cycle, Li Z et al. Nat Plants. 2016).

3. considering the peculiarities of the model species employed I'd be more cautious in driving general evolutionary hypothesis as in figure 4.

4. all results from this work originated from a genetic screening that is submitted elsewhere and is not available. There is thus an important piece of information missing.

5. DSR15-comp12 line showed only partial complementation while DSR28-comp13 line showed full complementation. This aspect has not been enough discussed. How many lines have been analyzed for each construct? Is the CrVenus-3xFLAG tag the reason behind this partial complementation? Did the authors try to complement DSR15 mutant with the full-length sequence of the candidate gene without tag?

Reviewer #3 (Remarks to the Author):

Photosynthetic organisms use photoprotection, a mechanism that protects the organism from damaging effect of excess light to optimize photosynthesis and survival. The manuscript "The CONSTANS flowering module in algal photosynthesis" by Tokutsu et. al. demonstrated that the primitive unicellular photosynthetic algae *Chlamydomonas reinhardtii* (Cr) protects itself from excess light using a novel transcriptional module consisting of CONSTANS/NUCLEAR FACTOR Y B/C (CrCO/NF-YB/C). This function of CO/NF-YB/C transcriptional complex is distinct from land plants where they control photoperiod-dependent flowering. From the evolutionary perspective in both the cases, the CO/NF-YB/C module is required for the reproductive success of the organism- i.e. increasing the population of *C. reinhardtii* and seed production in the land plants. This manuscript has clear genetic and molecular evidence to support the functional significance of CrCO/NF-YB/C in photoprotection. However, evidence to support the evolutionary perspective is incomplete and requires further study. Most of the arguments about the evolutionary significance of CrCO/NF-YB/C in this manuscript is discussed for CrCO by Serrano et. al. 2009. The evidence for the evolutionary role of NF-Y requires additional experimentation. I have following comments about the manuscript before recommending it for publication.

Major Comments

1. The CrCO complements the *Arabidopsis thaliana* (At) co mutant (Serrano et al 2009). I am curious to know if this evolutionary conserved transcriptional module (CrCO/NF-YB/C) binds to similar DNA elements (CORE elements) as in the land plants? Do these photoprotection genes have conserved promoter architecture?
2. What are the closest orthologs of CrNF-YB and CrNF-YC in *Arabidopsis*? Can the CrNF-YB and CrNF-YC rescue the late-flowering phenotype of *Arabidopsis* *nf-yb2 nf-yb3* and *nf-yc3 nf-yc4 nf-yc9* mutants?
3. Why does NF-YB show association with promoters of photoprotective genes under low light (Fig 2C and Extended data Fig 5)?
4. Unlike DSR28-comp28 the DSR15-comp12 has delayed response for maximum quantum yield of PSII (Fv/Fm) and reduced rescue for qE quenching capability under high light (Fig 1C and 1D). What is an explanation for this observation?
5. Line 136: What are the evidence presented in this manuscript to support the role of CrCO/NF-YB/C in the circadian clock?
6. The most relevant and recent article need to be discussed and cited in this manuscript are –
 - a) Nerina et al *The Plant Cell*, Vol. 29: 1516–1532, June 2017**
 - b) Nerina et al *Trends in Plant Science*, April 2018, Vol. 23, No. 4
 - c) Myers and Holt *Current Opinion in Plant Biology* 2018, 45:96–102

Minor Comments

1. It is not clear from the method section whether the high light tolerance experiments are done under constant temperature or not. A tenfold increase in light (low light vs high light 30:300 $\mu\text{mol photons/m}^2/\text{s}$) will heat up the system that will have a detrimental effect on the growth of cells. Also, I don't understand why the normal growth parameters are different 50 $\mu\text{mol photons/m}^2/\text{s}$ at 25°C (this manuscript) vs 40 $\mu\text{mol photons/m}^2/\text{s}$ at 23 °C (Ahira et. al. *Nature Plant*).
2. The title should be more intuitive photosynthesis should replace by photoprotection
3. Line 136 – There is no figure 5 – It should be the figure 4.

Reviewer #4 (Remarks to the Author):

This paper reports the characterization of new transcription factors (TFs) that are involved in regulating the photoprotective response in the model green algae, *Chlamydomonas reinhardtii*. Interestingly, the TFs described here are orthologues of genes known to control flowering time in angiosperms.

The TFs were identified from a genetic screen that sought to discover factors that control the expression of the photoprotective protein LHCSR (the manuscript describing this screen was included in the material for reviewers).

The reported experiments follow a logical order to show that:

- 1) The CONSTANS/NF-Y TFs are required for photoprotection and fitness in excess light using independent mutant lines and complemented strains.

- 2) That mutants in these TFs do not accumulate significant amount of transcripts of the LHCSR/PSBS genes and their corresponding protein products, which are required for nonphotochemical quenching (photoprotection).
- 3) That the CONSTANS/NF-YB/NF-YC proteins interact via a yeast two-hybrid assay. This is also the case for the Angiosperm complex . Immunoprecipitation experiments support these results.
- 4) ChIP experiments show that the TF complex binds to LHCSR/PSBS genes.

Furthermore, the manuscript continues to explore the differences and similarities between this TF complex between angiosperms and *Chlamydomonas*. This section is fairly dense and the non-expert would have difficulty following the description of the results relating the response of this TF complex to UV light. The manuscript would be improved by expanding this section (see minor issues below). The final figure of the manuscript illustrates a hypothesis about the divergence of CONSTANS/NF-YB/NF-YC function between angiosperms and green algae.

Overall, the experiments presented in this work are of high quality, with suitable replication and with appropriate statistical tests. The discovery of these regulatory elements are an important step towards understanding how photosynthetic organisms balance energy absorption vs dissipation in a natural, changing light environment. This work is likely to generate great interest within the in the field of photosynthesis, plant/algal physiology and stress regulation. It complements the physiology/biochemistry/genetic regulation studies previously reported in the field.

I have only a few comments that could improve the manuscript and I hope the authors find them helpful.

Major issue:

Early studies of LHCSR by Guertin's group showed the response of this transcript to some of the conditions reported here. The discussion should include these previous results. LHCSR was formerly called LI818, with LI standing for Light Induced transcript. PMID: 8980495 and 1371402 would be particularly germane for this discussion.

Minor issues:

I found the paragraph within lines 103-128 was difficult to read and interpret. I think this needs to be broken up into several different paragraphs to improve clarity.

Line 118. Stating the source of the *spa1* mutant at this point would aid in manuscript clarity (it is mentioned in the methods, but this is the first we encounter this in the main body of the paper).

Lin 274. Missing symbol in this pdf, perhaps should read degrees C?

Extended Data Figure 5 –The method being used here is qPCR, not qRT-PCR, which suggests reverse transcriptase was used.

Extended Data Table 5 refers to primers found in " Table S1" which doesn't exist in this manuscript.

Extended Data Table 5 lists multiple primer pairs used to investigate mRNA circadian rhythms. However, the only data presented in the manuscript appears to be given for the CrCO transcript (Extended Data Figure 8). Some changes in either data reporting or this table are required.

Methods :

Please expand on the "airbrush" cell disruption method.

Shevchenko et al. reference (IP and MS analysis section) is not in reference list.

End of comments.

We sincerely thank all the reviewers for taking the time to carefully review our manuscript. Please find our responses to your comments, suggestions, and questions below.

Reviewer #1 (Remarks to the Author):

1. Comments for Authors

Tokutsu and colleagues describe the identification of mutants affected in high light response in *Chlamydomonas reinhardtii* and the involvement and conservation of the plant CONSTANS homologue CrCO and plant NF-YB and NF-YC homologues in photoprotection. Employing biochemical and molecular biology tools they also identify a possible role for UV receptor UVR8 and E3 ubiquitin ligases homologues of COP1 and SPA1 in this response. They claim then that a conserved photoprotection signal involving this regulatory protein interaction network is conserved between algae and plants. The work is very interesting and adds important information to the field. However, the actual version raises serious concerns that need to be addressed in detail. For clarity reasons I will separate major from minor points.

Major points.

- The conclusion made by the authors that CrCO and CrNF-YB mutants affect only photoprotection genes is not completely correct. It has been described that CrCO mutation and overexpression affect cell division and starch metabolism in algae and indeed a severe CrCO reduction or overexpression caused extreme growth phenotypes in the recombinant *Chlamydomonas* strains in high light. Previous authors had exposed the algae to different photoperiods in minimal media and the lethal phenotype was explained as a severe lack of photoperiod response leading the algae to incorrect growth, division, reproduction and eventually death. If photoprotection is one more of these aspects the others cannot be ignored. In order to put the data into a general context, I suggest a more massive expression analysis should be carried out. Microarrays or RNAseq experiments between wild type and mutant strains (as well as overexpressor lines) should be carried out and the connection with photoprotection deduced from the total.

Thank you for the suggestions for conveying our findings in a broader context. In fact, we did not claim that the mutation in CrCO and/or NF-YB “only” affected the photoprotective response. The mutants defective in *CrCO* (*DSR10* and *DSR15*) and those defective in *NF-YB* (*DSR28* and *CC4286*) were originally identified as deficient in photoprotective responses (photoprotective gene expressions)(Ref #13; Tokutsu et al. Sci. Rep. (2019) 9: 2820). In this

study, our goal was to clarify the roles of CrCO and NF-YB in photoprotection of *C. reinhardtii*. The CHIP assays revealed that these two proteins (CrCO and NF-YB) were directly associated with the photoprotective genes, which occurred *in vivo* under high light (HL) conditions. We thus feel our current dataset is sufficient to achieve the goal of this study. The previous version of the manuscript was not clear regarding this fact, as the Reviewer pointed out. We amended the text to explicitly state it (Lines 95-97).

Nevertheless, we were happy to follow the suggestions of this reviewer in that the CrCO mutant could have an extreme growth phenotype because of its effects on cell division and starch metabolism, as reported in a previous study using the recombinant mutants (Serrano et al. 2009). However, our new experiments did not indicate a severe growth defect in our *crco* mutants, including *DSR10* and *DSR15* (Supplementary Fig. 4a), even under a light intensity of 100 $\mu\text{mol photons/m}^2/\text{s}$. Moreover, we searched for differences in cell shape that might reflect changes in the cell division process and starch metabolism (Serrano et al. 2009) without success (Supplementary Fig. 4b). Although we still cannot exclude the possibility that the photoperiodic response of the *CrCO* (and *NF-YB*) mutants is altered, our results imply that the mutation in *CrCO* (and *NF-YB*) has little effect on the cell division process under light conditions of 30-100 $\mu\text{mol photons/m}^2/\text{s}$. Currently, it is difficult to explain the discrepancy between the previous results and the results in this study, but as the reviewer kindly reminded us, it would be useful to include these additional data as Supplementary Information in the current manuscript. We also modified the main text to describe the growth phenotypes (Line 90-95). The massive expression analysis is a great idea to further extend the research, but it would obviously be beyond the scope of a single manuscript.

- The interactions between CrCO and NYs, CrCO and CrCOP1 need a deeper characterization than Yeast-two-hybrid data. I suggest co-localization, BiFC or FRET data in order to quantify the interactions. Co-IP in which the actual presence of the protein is demonstrated only for NF-YC-B with mass data is not enough as this is a very sensitive and not quantifiable process. CO-IP followed by Immunodetection is an effective and measurable method as well. No data other than Y2H is shown for CrCO-NFs and CrCO-CrCOP1/SPA1 interaction.

Thank you for the insightful comment and suggestion. It is technically challenging to quantify the interaction among the proteins by BiFC and/or FRET in the green alga because expression levels of different proteins are not practically controllable by current genetic techniques available for *C. reinhardtii*. Moreover, we were unfortunately unable to conduct Co-IP

experiments followed by immunodetection because we could not generate reliable antibodies against either CrCO or NF-Ys. To more deeply characterize the interactions between CrCO and NF-Ys, CrCO, and COP1, we included the Co-IP experiment of CrCO with full LC-MS/MS data in the revised manuscript (new Supplementary Table 1). The interaction between CrCO and NF-YB was additionally shown by the identification of NF-YB in the CrCO co-immunoprecipitate. Considering that both the co-immunoprecipitation (Supplementary Table 2) and yeast two-hybrid assays (Fig. 2b and S6) indicated that NF-YB interacts with both CrCO and NF-YC, it is probable that CrCO, NF-YB, and NF-YC form a complex. We added these data and revised the text accordingly (Lines 128–132).

If CrCO continuously degraded under low light (Fig. 3d and Supplementary Fig. 11b), detection of the CrCO/E3 ubiquitin ligase complex to evaluate their interaction would be extremely difficult. However, our LC-MS/MS analysis of CrCO–FLAG IP samples provided evidence that the CrCO–FLAG protein was associated with ubiquitinated protein(s) (Supplementary Table 5). We also adopted a genetic approach. CrCO and the photoprotective proteins were expressed even under low light conditions when the *spa1* mutation was introduced in the CrCO–Venus3xFLAG rescued background (*spa1 crco-2/CrCO*, Fig. 3d), indicating that the E3 ubiquitin ligase component SPA1 is clearly involved in the degradation of CrCO. These results imply that the E3-ubiquitin ligase complex is involved in CrCO degradation, possibly via direct interaction. We revised the text accordingly (Lines 186–196).

- Lines 123-128. I think this is a major assumption. The paper or the literature does not show any data that this is true and this cannot be supported by a model as suggested in line 125 ‘...appear to be conserved (Fig. 4), ...’. Even when I tend to agree with the authors that this might be true, still more consistent data should be shown to support this.

Thank you for the helpful comment. As also suggested by the other reviewers, we should eliminate over-speculation. In the revised manuscript, we completely rewrote this part (Lines 217-226). As for the hypothetical model, we focused on the experimental evidence obtained in this study, namely the central signaling pathways (SPA1/COP1-dependent E3 ubiquitin ligase to CONSTANS/NF-YB/NF-YC transcriptional complex) appear to be conserved between *A. thaliana* and *C. reinhardtii*. Figure 4 was modified accordingly.

- Complementation of CrCO and NF-YB mutants is achieved with the corresponding genes expressed under the control of the dual promoter RBs/Hsp70. This promoter is affected by

regular light (and probably UV light) as can actually be seen in the amount of transcripts in supplemental data (Fig S7a) induced by this promoter. No claim of continuous expression without the correct controls can be assumed by the authors. Use of a UBG or other constitutive promoters (or better its own promoters) should be tested.

As the reviewer reminded us, the activity of the HSP70A/RBCS dual promoter is moderately affected by light, and it is therefore not appropriate to state, “constitutive expression”. However, the HSP70A/RBCS dual promoter is one of the most widely used strong promoters to overexpress a gene of interest in *C. reinhardtii* (see, for instance, Onishi and Pringle (2016) *G3* 6:4115-4125; Kinoshita et al. (2017) *PLoS Genet.* 13: e1006645; López-Paz (2017) *Plant J.* 92: 1232-1244; Viola et al. (2019) *Plant J.* doi: 10.1111/tbj.14300), whose activity is high even under dark or low-light conditions (Schroda et al. (2000) *Plant J.* 21: 121-131), thereby suitable for complementation test. Therefore, we amended the text in the revised manuscript, as follows: “it overexpressed *CrCO* mRNA even before UV illumination” (Lines 179–180).

- Lines 135-136. While CO expression is controlled by the clock through GI/FKF1 action in *Arabidopsis*, in *Chlamydomonas* a clock-control of CrCO expression is suggested by its reported photoperiodic-constant light expression analysis reported previously only. However, no effect of CO or CrCO in the clock has been described so far. This should be corrected.

We apologize for the confusion. We did not claim that CrCO/NF-YB/NF-YC functions in the circadian clock. *CrCO* mRNA expression clearly showed a circadian rhythm (Supplemental Fig. 13). When considered that together with the function of the CrCO/NF-YB/NF-YC complex in the photoprotective response in *C. reinhardtii* (Figs. 1 and 2), it is plausible that *CrCO* connects the circadian clock signals and the photoprotective response in the green alga. The text was modified to clarify this point (Line 205).

- Materials and methods section and supplemental data FigS4 mention the edition of NF-YC gene and the phenotype resulting therein. Considering the importance of this experiment it is remarkable that no mention is made in the text. Authors should describe the figure, even consider adding as a main text, as well as describe and discuss these results in the text.

We appreciate very helpful comments. A detailed description of the *NF-YC* mutant phenotype was added in the revised manuscript (Lines 128–132).

- In the confocal images showing the nuclear localization of proteins no control of the nuclei

exists. Authors should use a dye or a known control to demonstrate that the actual distribution of CrCO, CNYs and CrCOP1/SPA1 are indeed imported into the nucleus. This is particularly important for the nuclear import of COP1 to the nucleus shown in Fig. 3a and S6.

Thank you for the constructive comment. We did not show the nuclear import of SPA1/COP1 in this study. We showed that UVR8 was translocated to the vicinity of the nucleus after UV illumination, and it then interacted with SPA1/COP1 (Fig. 3a and 3b). Therefore, we only described the behavior of UVR8 during UV illumination, using live-cell imaging and co-immunoprecipitation experiments.

It is certainly important to show the actual distribution of the proteins (CrCO, NF-YB, and UVR8). Therefore, we performed immunocytochemistry using the FLAG antibody and DAPI for visualizing Venus-3xFLAG fusion proteins and the nucleus, respectively (Supplementary Fig. 5). CrCO, NF-YB, and UVR8 all fused with Venus-3xFLAG, and as visualized by FLAG immunocytochemistry, were co-localized with DAPI signals under UV illumination. We therefore confirmed that CrCO, NF-YB, and UVR8 were indeed co-localized to the nucleus in *C. reinhardtii*. We added the immunocytochemistry results and descriptions in the revised manuscript (Supplementary Fig. 5 and Lines 125-126).

- The rhythmic expression of CrCO has been already demonstrated in previous literature even in LL conditions, authors should clarify why the figure S8 is needed at all.

The rhythmic expression pattern of *CrCO* in this study was different from that previously observed (Serrano et al. 2009, Fig. 1F). Serrano et al. observed expression peaks during both the light and dark periods, whereas we observed an expression increase only during the subjective daytime (Supplementary Fig. 13 [previous Supplementary Fig. 8]). This is why Figure S13 is needed. The increase in *CrCO* expression, especially during subjective daytime, would be beneficial for *C. reinhardtii* cells to efficiently drive the photoprotection mechanism under the light period in nature. We modified the text to clarify the significance of the results shown in Supplementary Fig. 13 (Lines 200–201).

Minor points.

- The title needs to be modified. The flowering module concerning CONSTANS, involve the actual CDF-CO-FT expression control, so that a hierarchical order seemed to be needed to detect and regulate the photoperiodic flowering response. This module is partially conserved in algae, where a homologous DOF transcription factor similar to CDFs (CrDOF) control CrCO

expression. However, the relationship described in this work cannot be considered a module because this is a protein complex and no hierarchy or “module” has been described. I would suggest a change in the title to something like ‘The CONSTANS flowering complex in algal photosynthesis’ or similar meaning.

Thank you for the comment. We changed the title to “The CONSTANS flowering complex controls the protective response of algal photosynthesis.”

- line 49, ‘..., these findings imply that the...’. This implication is not clear from the data presented in this work. I would rather use the term ‘suggest’.

Thank you for the comment. We modified the term to “suggest” (Line 71).

- In page 4, first lane, authors refer to a Fig 5, but this does not exist in the figure legends or in any actual picture sent with the manuscript. Please correct or delete.

Thank you very much for noticing this mistake. We corrected it as Fig. 4.

- Fig 4: in the model presented by the authors it is implicated that CrCO is ubiquitinated and degraded by the proteasome. This is only deduced by the interaction with CrCOP1 but there is no actual data to confirm this point, so it cannot be described as such in the model. It is also assumed that CR2 affect CrCO protein through CrCOP1/CrSPA1 but this is also an assumption as no such data has been still published to my knowledge in *Chlamydomonas*.

Thank you for the comments. To clarify whether CrCO degradation depends on proteasome activity, we tested the effect of the proteasome inhibitor MG132 on cells grown under low-light (LL) conditions. As a result, an increase in MG132 concentration clearly facilitated CrCO–FLAG protein accumulation even under LL conditions (Supplementary Fig. 12). Moreover, LC-MS/MS analysis of CrCO–FLAG IP samples provided evidence that a ubiquitin could be associated with the CrCO–FLAG protein (Supplementary Table 5). These results suggest that CrCO is degraded by proteasomes after ubiquitination by the E3 ubiquitin ligase in *C. reinhardtii*. In the revised manuscript, we presented these results and modified text to discuss the CrCO degradation (Lines 186–196).

We apologize that our hypothetical model caused confusion. Fig. 4 is to show a hypothetical model depicting possible signal transduction pathways in land plants (*A. thaliana*) and in *C. reinhardtii*. In the revised manuscript, we focused on the experimental evidence obtained in this study, namely the central signaling pathways (SPA1/COPI-dependent E3

ubiquitin ligase to CONSTANS/NF-YB/NF-YC transcriptional complex) appear to be conserved between *A. thaliana* and *C. reinhardtii*. Figure 4 was modified accordingly.

- In materials section, immunoblot description; protein concentration is measured by assuming 1 microgram of chlorophyll. While this could be a poorly assumable protein concentration measure for the same cells in the same repetitive conditions, this cannot be the standard for different cells with different recombinant elements and particularly when dealing with light-dependent processes where chlorophyll and protein content may differ enormously. Proteins should be measured by standard methods and described in each individual blot and figure.

Thank you for noticing this. The description in the previous manuscript was wrong. We used the same amount of protein samples for the immunoblots, which were extracted from 2×10^6 cells. We amended the “Immunoblot analysis” section in the Methods submitted to “Protocol Exchange”.

- Immunoprecipitation experiments are not described in sufficient detail. i.e. ‘Cells treated with light’ means standard light, UV light, low light, blue light? Buffer for protein solubilization (other than in the presence of triton X-100) is not described.

We provided more detailed information regarding our immunoprecipitation method in the revised Methods (submitted to “Protocol Exchange”).

- For protein gels, corresponding size and molecular mass standards should be shown in the pictures. Sometimes more than 1 band are shown and this is not clarified in the text.

We actually presented the molecular mass standards in the protein gel photograph (CBB-stained protein gel; Fig. 3b). However, as the reviewer mentioned, we did not describe the molecular mass standards in the legend. We amended the figures and the legends, accordingly. Further, we added the molecular mass information to the immunoblot photographs in the revised figures (Figs. 1, 3, and Supplementary Fig. 6).

- CrCOP1 and CrSPA1 cloning is not fully described in the text. Size of the amplified fragments, the way they were isolated (or borrowed from another laboratory) should be described or acknowledged in the text.

Thank you for the suggestion. We modified the methods to include more detailed information including the amplified fragments used for the yeast two-hybrid assay (Supplementary Table 8).

- Negative controls for CrCO or CrNF-YB ChIP experiments are not described. A promoter where the binding of CO/NF complex is not light dependent or a promoter where the complex does not bind should be tested as well.

In the present version of our manuscript, we included negative controls for the ChIP experiments (Labeled as *CBLP* control or FLAG control). We do not know of a promoter that binds to the CrCO/NF-Y complex in a manner that does not depend on light in *C. reinhardtii*. Therefore, we used the *CBLP* gene, which is not a light-responsive gene in this green alga, as the negative control for the ChIP-PCR analysis. Moreover, we used the *LHCSR1-Luc717* strain transformed with the Venus-3xFLAG construct without CrCO or NF-YB as the negative control for our ChIP analysis because the Venus-3xFLAG protein should not specifically associate with the promoter where CrCO and/or NF-YB binds.

- Supplemental data is referred in some sentences as ‘extended data Fig. n’, while in others, the same picture is called ‘Fig. Sn’, please correct.

Thank you for noticing these mistakes. We have corrected them in “Supplementary Fig. n”.

- mRNA methods. How much RNA was used to make cDNA to test for gene expression? Was it always the same quantity?

We used the same amount of RNA (300 ng) for cDNA generation. We modified the Methods (submitted to “Protocol Exchange”) accordingly to describe the amount of RNA used for the experiments.

Reviewer #2 (Remarks to the Author):

The manuscript “The CONSTANS flowering module in algal photosynthesis” by Tokutsu, Fujimura-Kamada, Matsuo, Yamasaki, and Minagawa deals with the evolution of the transcriptional complex that regulates flowering in land plants. A large number of experiments has been performed, the results are in general robust and clearly presented.

1. I have one major concern with the general significance of the work. qE response in

Chlamydomonas reinhardtii is induced when cells are exposed to strong illumination that induce the accumulation of LHCSR/PSBS. This is not true for most plants and eukaryotic algae where qE is constitutively present. Now that more data on different species are available it is clear that *Chlamydomonas* has a peculiar behavior in this respect. There is no evidence to my knowledge that signaling pathway inducing qE is conserved in other photosynthetic organisms. It is thus possible that results presented here are only valid for this species, questioning the general significance of the work.

Thank you for the comment. It is correct that qE is constitutively present in land plants. The crucial qE effector PsbS is constitutively expressed in land plants. This is probably because qE is constantly required on land due to the UV environment. It has been generally discussed that the inducible qE system developed in aquatic organisms (eukaryotic algae) was lost once they landed a long time ago (when they became plants). The inducible qE system is actually common among eukaryotic algae. For instance, another model alga, a diatom, *Phaeodactylum tricorutum*, also shows light inducible accumulation of LHCXs, homologs of LHCSRs (Taddei et al. 2016 J. Exp. Bot.). Because the molecular mechanisms of photoprotection in the other algae, however, have not been investigated at a sufficient depth as compared to those of *C. reinhardtii*, we should not conclude that *C. reinhardtii* has a peculiar photoprotective system among photosynthetic organisms. We, not only us, rather assume that light-inducible qE is a common tactic in aquatic photosynthesis. In light of the current knowledge and based on our own study, we hypothesize that the qE induction mechanism in *C. reinhardtii* is possibly applicable for other algae. It would be fascinating if other researchers tested this hypothesis in other organisms.

2. In all manuscript data obtained with *Chlamydomonas* are referred as valid for all green algae. see as example lines 70-71: “In green algae, qE-type photoprotection is a light-inducible process that involves the expression of photoprotective proteins^{19,20}.”. in particular for qE and regulation of photosynthesis we know since a few years that *Chlamydomonas* has peculiar behavior (as example see the xanthophyll cycle, Li Z et al. Nat Plants. 2016).

Thank you for the comment. We agree with the reviewer’s comment and have modified the main text and hypothetical model so as to not overgeneralize *C. reinhardtii* as a typical green alga.

3. considering the peculiarities of the model species employed I’d be more cautious in driving

general evolutionary hypothesis as in figure 4.

As also suggested by the other reviewers, we agree that we need to reconsider the hypothetical model more carefully. As other green algae, for instance a primitive green alga (prasinophyte) *Ostreococcus tauri*, has *CO* (Valverde 2011, J. Exp. Bot.) and *NF-Y* (Thiriet-Rupert et al. 2016, BMC Genomics), it is conceivable that the *CO/NF-Y* regulatory system is possibly widely conserved in green algae. In the present research, we found that the light-dependent signal transduction kernel, “COP1/SPA1 to *CO/NF-Y*,” is common between *C. reinhardtii* (Chlorophyta) and *A. thaliana* (Streptophyta), which suggests a possibility that this light-dependent signal transduction pathway may have been established in the common ancestral organism(s). Considering that such a “kernel” controls different downstream genes between *A. thaliana* (*FT*) and *C. reinhardtii* (*LHCSR*s and *PSBS*), it is plausible that the downstream target genes and physiological functions may have diversified afterward. Based of the above discussion, we modified the hypothetical model that focuses on the comparison of the signal transduction pathways between *A. thaliana* and *C. reinhardtii* (revised Figure 4). We also modified the main text to explicitly state this as a hypothesis and list literature supporting the hypothesis (Lines 217–226).

4. all results from this work originated from a genetic screening that is submitted elsewhere and is not available. There is thus an important piece of information missing.

We apologize for this inconvenience. Meanwhile, the screening paper was published (Tokutsu et al. Sci. Rep. (2019) 9: 2820), and is now cited in the revised manuscript (Ref. 9).

5. DSR15-comp12 line showed only partial complementation while DSR28-comp13 line showed full complementation. This aspect has not been enough discussed. How many lines have been analyzed for each construct? Is the CrVenus-3xFLAG tag the reason behind this partial complementation? Did the authors try to complement DSR15 mutant with the full-length sequence of the candidate gene without tag?

Even though we screened more than 5,000 selective antibiotic-resistant clones, we only found several clones that showed LHCSR1 protein accumulation after UV illumination. Among those, only the *crco-2/CrCO* strain (previous naming, *DSR15-comp12*) showed clear CrCO-FLAG protein accumulation as described in the main text (Fig. 1 and Supplementary Fig. 11b). However, we found that the *crco-2/CrCO* strain showed only partial complementation of photoprotective responses (Fig. 1). This observation led us to assume that the Venus-3xFLAG

tag in the C-terminus of CrCO might affect the precise accumulation and/or localization of CrCO in *C. reinhardtii* (Lines 113–119).

On the other hand, in this study, our goal was to clarify whether CrCO accumulation rescues the photoprotection defect in *C. reinhardtii*. The HL-tolerant assay revealed that the CrCO–FLAG accumulation in the *crco-2/CrCO* strain was positively correlated with expression of photoprotective genes, which could be essential for photoprotection under high light (HL) conditions (Fig. 1). We therefore used the *crco-2/CrCO* strain as a rescued strain of the *CrCO* mutant in this study.

Reviewer #3 (Remarks to the Author):

Photosynthetic organisms use photoprotection, a mechanism that protects the organism from damaging effect of excess light to optimize photosynthesis and survival. The manuscript “The CONSTANS flowering module in algal photosynthesis” by Tokutsu et. al. demonstrated that the primitive unicellular photosynthetic algae *Chlamydomonas reinhardtii* (Cr) protects itself from excess light using a novel transcriptional module consisting of CONSTANS/NUCLEAR FACTOR Y B/C (CrCO/NF-YB/C). This function of CO/NF-YB/C transcriptional complex is distinct from land plants where they control photoperiod-dependent flowering. From the evolutionary perspective in both the cases, the CO/NF-YB/C module is required for the reproductive success of the organism- i.e. increasing the population of *C. reinhardtii* and seed production in the land plants. This manuscript has clear genetic and molecular evidence to support the functional significance of CrCO/NF-YB/C in photoprotection. However, evidence to support the evolutionary perspective is incomplete and requires further study. Most of the arguments about the evolutionary significance of CrCO/NF-YB/C in this manuscript is discussed for CrCO by Serrano et. al. 2009. The evidence for the evolutionary role of NF-Y requires additional experimentation. I have following comments about the manuscript before recommending it for publication.

Major Comments

1. The CrCO complements the *Arabidopsis thaliana* (At) *co* mutant (Serrano et al 2009). I am curious to know if this evolutionary conserved transcriptional module (CrCO/NF-YB/C) binds to similar DNA elements (CORE elements) as in the land plants? Do these photoprotection genes have conserved promoter architecture?

In this study, our goal was to elucidate the association of the CrCO/NF-Y complex with the promoter region of the photoprotective genes. The ChIP assays revealed that both CrCO and NF-YB were directly associated with the photoprotective genes (Fig. 2c and Supplementary Fig. 9). We did not reveal whether this association was mediated by DNA elements (CORE elements), as in *A. thaliana*. To address the reviewer's question, we searched for and visualized the sequences of CORE elements and *cis*-elements for CONSTANS and NF-Y, respectively (Supplementary Fig. 8). We found that the elements were widely located in the upstream promoter region of the representative photoprotective genes. This result, together with the results of the ChIP-PCR assays, implies that the CrCO/NF-YB/NF-YC transcriptional module possibly binds to similar DNA elements as in land plants. Accordingly, we modified the manuscript to discuss the DNA elements in the photoprotective genes (Lines 144–147).

2. What are the closest orthologs of CrNF-YB and CrNF-YC in Arabidopsis? Can the CrNF-YB and CrNF-YC rescue the late-flowering phenotype of Arabidopsis *nf-yb2 nf-yb3* and *nf-yc3 nf-yc4 nf-yc9* mutants?

Thank you for the constructive comment. We performed a phylogenetic analysis of NF-YB and NF-YC in both *C. reinhardtii* and *A. thaliana* (Supplementary Fig. 1). The resulting phylogenetic tree shows that CrNF-YB and CrNF-YC are clustered with AtNF-YB1/8/10 and AtNF-YC1/2/3/4/9, respectively. In the future, we would like to evaluate this phylogenetic relationship by testing whether CrNF-YB and CrNF-YC can rescue *Arabidopsis* NF-Y mutants. In the present research, we revised the manuscript to include the results of the phylogenetic analysis (Lines 68–70).

3. Why does NF-YB show association with promoters of photoprotective genes under low light (Fig 2C and Extended data Fig 5)?

Thank you for the question. We were also curious about this phenomenon. Previous structural research of the NF-Y complex revealed that the NF-YB/NF-YC complex binds to the sugar-phosphate backbone flanking the core element of NF-YA (CCAAT) and facilitates association of NF-YA with the genome (Nardini et al. 2003 Cell). Moreover, extensive plant research on NF-Y and CONSTANS has suggested that the NF-YB/NF-YC complex is also associated with CONSTANS (CCACA binding). Therefore, the NF-YB/NF-YC complex is suggested to bind near those DNA elements (CCAAT or CCACA) to support the association of either NF-YA or CONSTANS with the genome (Myers and Holt 2018 Curr. Opin. Plant Biol.).

Based on this feature of the NF-YB/NF-YC complex in land plants, we assume that NF-YB could weakly associate with the promoter region of photoprotective genes under LL in *C. reinhardtii*. This association is likely reinforced by the accumulation of CrCO and the formation of the CrCO/NF-YB/NF-YC complex under HL conditions. In the revised manuscript, we modified the text to describe the association of NF-YB in more detail (Lines 152–159).

4. Unlike DSR28-comp28 the DSR15-comp12 has delayed response for maximum quantum yield of PSII (Fv/Fm) and reduced rescue for qE quenching capability under high light (Fig 1C and 1D). What is an explanation for this observation?

The phenomena pointed by the reviewer is due to the partial (reduced) expression of the photoprotective proteins (LHCSRs and PSBS) in the *crco-2/CrCO* strain (previous naming, *DSR15-comp12* strain) under HL. For details, please read our response to Reviewer #2's Major comment #5. Furthermore, we discussed the importance of precise regulation of CrCO protein accumulation for photoprotective responses under HL conditions (Lines 213–216).

5. Line 136: What are the evidence presented in this manuscript to support the role of CrCO/NF-YB/C in the circadian clock?

The same point was raised by Reviewer #1 (Major comment #5). We have modified the text to clarify this point (Line 205).

6. The most relevant and recent article need to be discussed and cited in this manuscript are –

- a) Nerina et al The Plant Cell, Vol. 29: 1516–1532, June 2017**
- b) Nerina et al Trends in Plant Science, April 2018, Vol. 23, No. 4
- c) Myers and Holt Current Opinion in Plant Biology 2018, 45:96–102

Thank you for providing the useful information. We found that the information is very important for interpreting and discussing the promoter elements of CONSTANS and NF-Ys in *C. reinhardtii*. In the revised version of our manuscript, we cited the articles (Ref. 22 and 23) and discussed the promoter elements in the main text (Lines 144–147, 152–159).

Minor Comments

1. It is not clear from the method section whether the high light tolerance experiments are done under constant temperature or not. A tenfold increase in light (low light vs high light 30:300 $\mu\text{mol photons/m}^2/\text{s}$) will heat up the system that will have a detrimental effect on the growth of

cells. Also, I don't understand why the normal growth parameters are different 50 $\mu\text{mol photons/m}^2/\text{s}$ at 25°C (this manuscript) vs 40 $\mu\text{mol photons/m}^2/\text{s}$ at 23 °C (Ahira et. al. Nature Plant).

The experiments were performed under a controlled constant temperature of 25°C. We added this information to the Methods section. The differences in normal growth parameters were caused by differences in the control strains used in different studies. In this study, a common *C. reinhardtii* WT strain (137c) was used as a control strain, whereas the phototropin mutant (*phot*) was used as a control strain in the previous study (Aihara et al. 2019 Nature Plants). Because the *phot* strain is sensitive to HL (Petroutsos et al. 2016 Nature), a relatively weak light intensity was given in this study even for normal growth conditions. However, these differences were relatively small, 10 $\mu\text{mol photons/m}^2/\text{s}$ and 2°C; consequently, there should be no substantial differences in the results.

2. The title should be more intuitive photosynthesis should replace by photoprotection

Thank you for the comment. We changed the title to “The CONSTANS flowering complex controls the protective response of algal photosynthesis.”

3. Line 136 – There is no figure 5 – It should be the figure 4.

Thank you for noticing this error. We have corrected the mistake.

Reviewer #4 (Remarks to the Author):

This paper reports the characterization of new transcription factors (TFs) that are involved in regulating the photoprotective response in the model green algae, *Chlamydomonas reinhardtii*. Interestingly, the TFs described here are orthologues of genes known to control flowering time in angiosperms. The TFs were identified from a genetic screen that sought to discover factors that control the expression of the photoprotective protein LHCSR (the manuscript describing this screen was included in the material for reviewers). The reported experiments follow a logical order to show that:

- 1) The CONSTANS/NF-Y TFs are required for photoprotection and fitness in excess light using independent mutant lines and complemented strains.
- 2) That mutants in these TFs do not accumulate significant amount of transcripts of the LHCSR/PSBS genes and their corresponding protein products, which are required for nonphotochemical quenching (photoprotection).

3) That the CONSTANS/NF-YB/NF-YC proteins interact via a yeast two-hybrid assay. This is also the case for the Angiosperm complex . Immunoprecipitation experiments support these results.

4) ChIP experiments show that the TF complex binds to LHCSR/PSBS genes.

Furthermore, the manuscript continues to explore the differences and similarities between this TF complex between angiosperms and *Chlamydomonas*. This section is fairly dense and the non-expert would have difficulty following the description of the results relating the response of this TF complex to UV light. The manuscript would be improved by expanding this section (see minor issues below). The final figure of the manuscript illustrates a hypothesis about the divergence of CONSTANS/NF-YB/NF-YC function between angiosperms and green algae.

Overall, the experiments presented in this work are of high quality, with suitable replication and with appropriate statistical tests. The discovery of these regulatory elements are an important step towards understanding how photosynthetic organisms balance energy absorption vs dissipation in a natural, changing light environment. This work is likely to generate great interest within the in the field of photosynthesis, plant/algal physiology and stress regulation. It complements the physiology/biochemistry/genetic regulation studies previously reported in the field.

I have only a few comments that could improve the manuscript and I hope the authors find them helpful.

Major issue:

Early studies of LHCSR by Guertin's group showed the response of this transcript to some of the conditions reported here. The discussion should include these previous results. LHCSR was formerly called LI818, with LI standing for Light Induced transcript. PMID: 8980495 and 1371402 would be particularly germane for this discussion.

Thank you for the constructive comment. As the reviewer mentioned, Guertin's group reported circadian oscillation of *LI818* RNA, which is in fact compatible with the idea that CrCO/NF-YB/NF-YC-dependent regulation of the photoprotective genes is possibly under circadian clock control. Accordingly, we modified the main text to discuss this phenomenon, with citations from the previous studies as suggested by the reviewer (Ref. 31 and 32, Lines 202–204).

Minor issues:

I found the paragraph within lines 103-128 was difficult to read and interpret. I think this needs to be broken up into several different paragraphs to improve clarity.

Thank you for the suggestion. We have separated the paragraph into three paragraphs and tried to modify the text to improve clarity (Lines 163–175, 176–182, 183–196).

Line 118. Stating the source of the *spa1* mutant at this point would aid in manuscript clarity (it is mentioned in the methods, but this is the first we encounter this in the main body of the paper).

Thank you for the suggestion. We added information about the *spa1* mutant in the revised manuscript (Lines 183–186).

Lin 274. Missing symbol in this pdf, perhaps should read degrees C?

Thank you for noticing this mistake. We have corrected it.

Extended Data Figure 5 –The method being used here is qPCR, not qRT-PCR, which suggests reverse transcriptase was used.

Thank you for noticing this mistake. We have corrected it.

Extended Data Table 5 refers to primers found in “ Table S1” which doesn’t exist in this manuscript.

Thank you for noticing this mistake. We have corrected it.

Extended Data Table 5 lists multiple primer pairs used to investigate mRNA circadian rhythms. However, the only data presented in the manuscript appears to be given for the CrCO transcript (Extended Data Figure 8). Some changes in either data reporting or this table are required.

Thank you for noticing this mistake. We have corrected it.

Methods :

Please expand on the “airbrush” cell disruption method.

More information with a reference (Nishimura et al. 2002 Gene. Dev.) regarding the airbrush method was provided in the Methods submitted to “Protocol Exchange”.

Shevchenko et al. reference (IP and MS analysis section) is not in reference list.

Thank you for the input. We added this reference (Ref. 35).

Reviewers' comments:

Reviewer #2 (Remarks to the Author):

Authors answer to main point raised by rev#2.

I am sorry to say this, but I must say that did not address at all the major point raised and simply avoid the concern in their answer. Even in their answer authors are still assuming results from *Chlamydomonas* as valid for all eukaryotic algae. This is just wrong, there is a huge amount of data showing that algae biodiversity is very large (far larger than plants, as example). I'd strongly recommend authors to look to the literature to present their results in a more appropriate context.

To be more specific:

1. A constitutive qE is not only typical of land plants, but also of most eukaryotic algae. this is not a behavior that depends on the NPQ activator (PSBS or LHCSR) but rather if this activator is accumulated or not. *Chlamydomonas* is the exception rather than the norm.
2. that cited paper (as several others) showed that in diatoms LHCX genes are regulated and can increase their expression in response to environmental stimuli. But LHCX proteins are still significantly accumulated in all conditions and, in fact, cells show a strong constitutive NPQ response. This is the case for many different algae species but not for *Chlamydomonas*, that is instead peculiar since NPQ is normally negligible and only induced in HL.

Reviewer #3 (Remarks to the Author):

The revised version of the manuscript has addressed all of my questions and comments. I fully recommend accepting this manuscript for publication in Nature Communications.

Reviewers' comments:

Reviewer #2 (Remarks to the Author):

Authors answer to main point raised by rev#2.

I am sorry to say this, but I must say that did not address at all the major point raised and simply avoid the concern in their answer. Even in their answer authors are still assuming results from *Chlamydomonas* as valid for all eukaryotic algae. This is just wrong, there is a huge amount of data showing that algae biodiversity is very large (far larger than plants, as example). I'd strongly recommend authors to look to the literature to present their results in a more appropriate context.

We completely agree with Reviewer-2 such that ones are not supposed to assume results from *Chlamydomonas* as valid for all eukaryotic algae. There is a huge amount of data showing that algae biodiversity is very large. We took this criticism very seriously and gave a deep thought about how this misunderstanding has arisen.

To be more specific:

1. A constitutive qE is not only typical of land plants, but also of most eukaryotic algae. this is not a behavior that depends on the NPQ activator (PSBS or LHCSR) but rather if this activator is accumulated or not. *Chlamydomonas* is the exception rather than the norm.

Here, the reviewer probably means the NPQ activators (PSBS/LHCSR) are constantly accumulated in “most” eukaryotic algae. We are sorry, but we are not supposed to make a general statement like this at this point.

PSBS is only present in green plants (land plants, moss, ..., green algae), and when we think about algal PSBS, it is mainly studied in *Chlamydomonas*. In *Chlamydomonas*, we know it is light-inducible (if it contains UV) (Allorent et al. PNAS, 2016). Another study was done in *Ulva*, where PSBS is also light-inducible (Mou et al. Plant Biol., 2013).

LHCSR/LHCXs are widely distributed among eukaryotic algae. However, only those in green algae (*Chlamydomonas* and *Ulva*) and diatoms (*Phaeodactylum* and *Thalassiosira*) are studied at a sufficient depth as far as we know; In *C. reinhardtii*, LHCSR1/3 are light-inducible (Peers et al, Nature 2009; Allorent et al. PNAS, 2016). In *Ulva*, LHCSR is light-inducible (Mou et al. Plant Biol., 2013), In *P. tricornutum*, *Lhcx1* is marginally light-inducible (x2), but *Lhcx2/3* are highly light-inducible, and *Lhcx4* is dark-inducible (Taddei et al., 2016, J Exp Bot). In *T. pseudonana*, *Lhcx1* is marginally light-inducible (x2), but *Lhcx4/6* are highly light-inducible (Zhu and Green, BBA, 2010).

What we learn from here is that there are several light-inducible NPQ activator genes in eukaryotic algae. We can therefore simply hypothesize that light-induction of NPQ activator genes and the similar E3-ligase/CO-mediated mechanisms may be present in other algae. We would be, however, open to cite literatures, if any, describing that PSBS/LHCSR are constitutively expressed in “most” eukaryotic algae.

2. that cited paper (as several others) showed that in diatoms LHCX genes are regulated and can increase their expression in response to environmental stimuli. But LHCX proteins are still significantly accumulated in all conditions and, in fact, cells show a strong constitutive NPQ response. This is the case for many different algae species but not for *Chlamydomonas*, that is instead peculiar since NPQ is normally negligible and only induced in HL.

The current manuscript is focusing on the regulatory mechanism of NPQ activator genes in *C. reinhardtii*. In the (previous) revised manuscript, we paid special attention for keeping the description specific to *C. reinhardtii* thanks to this reviewer's comments. In the current version, we

put "... in the green alga *Chlamydomonas*" in the title as well and added the following sentence at the end of Discussion to explain the possible background argument (Lines 236-239).

"Whether we can see the same type of regulation in other algae would be a matter of future study as our current knowledge about their photoprotective behavior as well as the related genetic information is still very limited"

Because peculiarity/normality is a subjective matter, we are not going into it.

Reviewer #3 (Remarks to the Author):

The revised version of the manuscript has addressed all of my questions and comments. I fully recommend accepting this manuscript for publication in Nature Communications.

Thank you for your constructive comments on the previous manuscript. We are confident that all the comments are very useful and substantially improved our manuscript.